# Bidirectional Wnt signaling between endoderm and mesoderm confers tracheal identity in mouse and human cells

Keishi Kishimoto [1,2,3,4], Kana T. Furukawa [1], Agustin Luz-Madrigal[3,4], Akira Yamaoka [1], Chisa Matsuoka[1], Masanobu Habu[5], Cantas Alev [5,6], Aaron M. Zorn [2,3,4] & Mitsuru Morimoto [1,2 ✉]

The periodic cartilage and smooth muscle structures in mammalian trachea are derived from tracheal mesoderm, and tracheal malformations result in serious respiratory defects in neonates. Here we show that canonical Wnt signaling in mesoderm is critical to confer trachea mesenchymal identity in human and mouse. At the initiation of tracheal development, endoderm begins to express *Nkx2.1*, and then mesoderm expresses the *Tbx4* gene. Loss of *β-catenin* in fetal mouse mesoderm causes loss of Tbx4+ tracheal mesoderm and tracheal cartilage agenesis. The mesenchymal *Tbx4* expression relies on endodermal Wnt activation and Wnt ligand secretion but is independent of known *Nkx2.1*-mediated respiratory development, suggesting that bidirectional Wnt signaling between endoderm and mesoderm promotes trachea development. Activating Wnt, Bmp signaling in mouse embryonic stem cell (ESC)-derived lateral plate mesoderm (LPM) generates tracheal mesoderm containing chondrocytes and smooth muscle cells. For human ESC-derived LPM, SHH activation is required along with WNT to generate proper tracheal mesoderm. Together, these findings may contribute to developing applications for human tracheal tissue repair.

[1] Laboratory for Lung Development and Regeneration, Riken Center for Biosystems Dynamics Research (BDR), Kobe 650-0047, Japan. [2] RIKEN BDR—CuSTOM Joint Laboratory, Cincinnati Children's Hospital Medical Center, Cincinnati, OH 45229, USA. [3] Center for Stem Cell & Organoid Medicine (CuSTOM), Cincinnati Children's Hospital Medical Center, Cincinnati, OH 45229, USA. [4] Division of Developmental Biology, Cincinnati Children's Hospital Medical Center, Cincinnati, OH 45229, USA. [5] Department of Cell Growth and Differentiation, Center for iPS Cell Research and Application (CiRA), Kyoto University, Kyoto 606-8507, Japan. [6] Institute for the Advanced Study of Human Biology (ASHBi), Kyoto University, Kyoto 606-8501, Japan. ✉email: mitsuru.morimoto@riken.jp

The mammalian respiratory system is crucial for postnatal survival, and defects in the development of the respiratory system cause life-threatening diseases in breathing at birth[1]. The trachea is a large tubular air path that delivers external air to the lung. Abnormal development of the tracheal mesenchyme, including cartilage and smooth muscle (SM), is associated with congenital defects in cartilage and SM such as trachea–esophageal fistula (TEF) and tracheal agenesis (TA)[2,3]. Thus, understanding trachea development is crucial to better understand TEF/TA and establish a protocol to reconstruct trachea from pluripotent stem cells for human tissue repair.

Trachea/lung organogenesis is coordinated by endodermal-mesodermal interactions during embryogenesis. The primordial tracheal/lung endoderm appears at the ventral side of the anterior foregut at embryonic day 9–9.5 (E9.0–9.5) in mouse (Fig. 1a). Previous studies have demonstrated that development of tracheal/lung endoderm is initiated by graduated expression of mesodermal *Wnt2/2b* and *Bmp4* along the dorsal-ventral axis[4–7]. This mesodermal-to-endodermal Wnt and Bmp signaling drives expression of *Nkx2.1*, the key transcription factor of tracheal/lung lineage[8], at the ventral side of the anterior foregut endoderm, which in turn suppresses *Sox2* to segregate these Nkx2.1+ endodermal cells from the esophageal lineage. The Nkx2.1+ endoderm then invaginates into the ventral mesoderm to form the primordial trachea and lung buds. At the same time, the Sox2+ endoderm at the dorsal side develops into the esophagus by E10.5 (Fig. 1a)[9]. These studies have demonstrated that mesodermal cells secrete growth factors (e.g., Wnt and Bmp) to induce respiratory endoderm identity[4–6].

The tracheal mesoderm originates in the ventral fold of lateral plate mesoderm (LPM) surrounding the anterior foregut endoderm. After induction of endodermal *Nkx2.1* expression at E9.5, tracheal/lung mesoderm is defined by *Tbx4/5* at E10.5, which are markers for tracheal/lung mesoderm and required for proper mesenchymal development (Fig. 1a)[10]. In contrast to *Tbx5* which is also expressed in LPM and cardiac mesoderm[11,12], *Tbx4* expression is restricted to respiratory tissue. At E9.5, *Tbx4* is only detected in lung bud mesoderm but not tracheal mesoderm (Supplementary Fig. 1). *Tbx4* expression is then detected in tracheal mesoderm from E10.5. *Tbx4* and *Tbx5* cooperate to steer normal trachea development. Both genes are required for mesodermal development of the trachea, particularly for cartilage and smooth muscle differentiation as well as morphogenesis. The crucial functions of these genes are validated by *Tbx4, 5* double mutants exhibiting the phenotypes of tracheal stenosis[10]. We previously reported that synchronized polarization of mesodermal cells and temporal initiation of cartilage development regulate tracheal tube morphogenesis by coordinating the length and diameter of the mouse trachea, respectively[13,14]. However, the mechanism underlying the initial induction of tracheal mesoderm is still unclear.

Here, we propose that this communication is bidirectional between endoderm and mesoderm. In our model, once tracheal endoderm is specified around E9.5, endodermal cells express Wnt ligands to induce *Tbx4* expression in tracheal mesoderm after E10.5. To substantiate the model, we address the following key issues: (1) tracheal endoderm secretes Wnt ligands; (2) tracheal mesoderm responds to endodermal Wnt ligands to specify

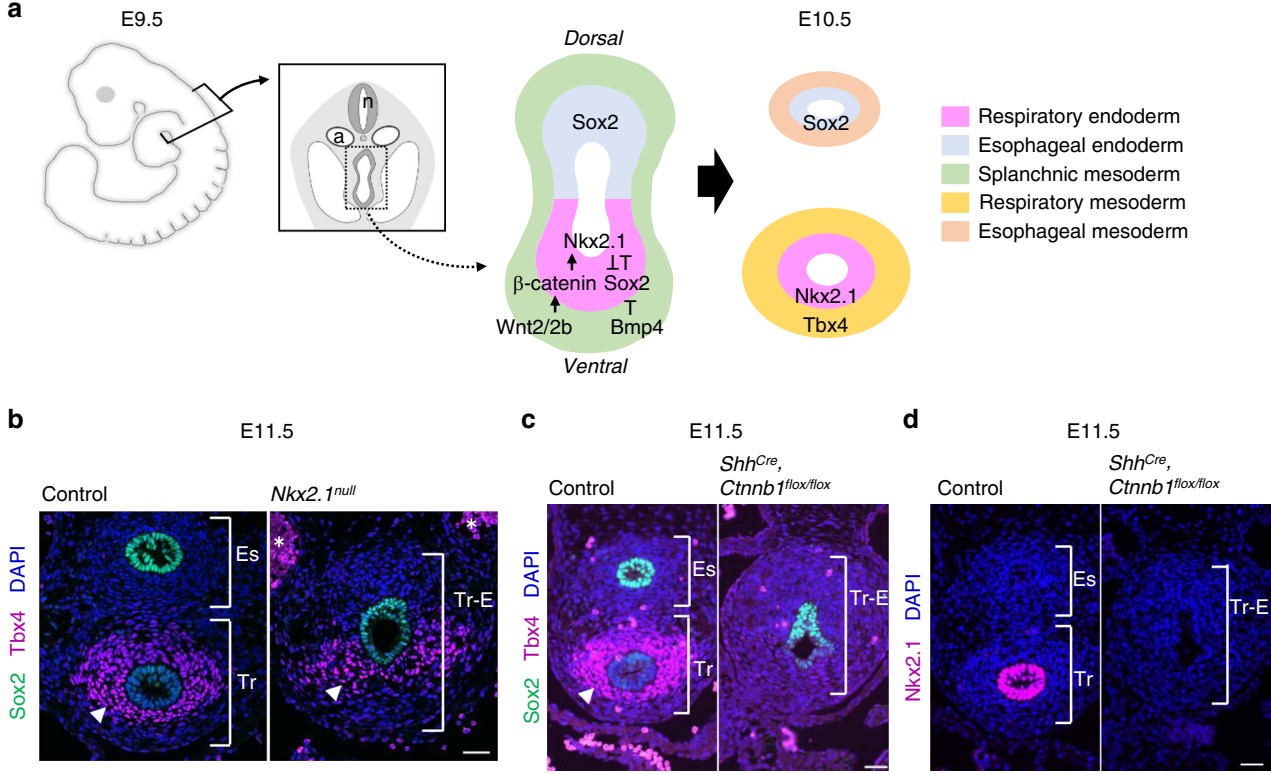

**Fig. 1 Activation of Wnt signaling in endoderm, but not *Nkx2.1* expression, is activated to promote mesodermal development of the mouse trachea.** **a** Schematic model of tracheoesophageal segregation. **b** Transverse sections of *Nkx2.1null* mouse embryos and littermate controls. Sections were stained for Sox2 (*green*), Tbx4 (*magenta*), and DAPI (*blue*). Arrowheads indicate Tbx4+ tracheal mesoderm. Asterisks indicate nonspecific background signal of blood cells in dorsal aorta. n = 3/3 embryos per genotype. **c** Transverse sections of *Shh^Cre, Ctnnb1^flox/flox* mouse embryos and littermate controls. Sections were stained by Sox2 (*green*), Tbx4 (*magenta*), and DAPI (*blue*). Arrowheads indicate Tbx4+ tracheal mesoderm. n = 3/3 embryos per genotype. **d** Transverse sections of *Shh^Cre, Ctnnb1^flox/flox* mouse embryos-, and littermate controls. Sections were stained by Nkx2.1 (*magenta*) and DAPI (*blue*). n = 3/3 embryos per genotype. n neural tube, a aorta, Es Esophagus, Tr Trachea, Tr–E Trachea–Esophageal tube. Scale bar, 40 μm.

mesodermal identity through *Tbx4* expression; (3) *Tbx4* is a direct Wnt target gene.

## Results

**Endodermal Wnt activity but not *Nkx2.1* initiates *Tbx4* expression in mouse tracheal mesoderm.** To study the initiation of the mesodermal development of the trachea, we validated the involvement of *Nkx2.1* in mesodermal *Tbx4* expression because endodermal-mesodermal interactions orchestrate organogenesis throughout development in general. *Nkx2.1* is an endodermal transcription factor necessary for tracheal and lung development and its genetic ablation results in TEF[8]. We examined *Nkx2.1null* mouse embryos and confirmed the TEF phenotype with a single trachea–esophageal (Tr–E) tube (Fig. 1b). Interestingly, *Nkx2.1null* embryos retained *Tbx4* expression in the ventrolateral mesoderm of a single Tr–E tube, although the segregation was defective (Fig. 1b), indicating that mesodermal induction of the trachea is independent of endodermal *Nkx2.1*. We compared the phenotype of *Nkx2.1null* with that of *ShhCre, Ctnnb1flox/flox* embryos, which also show anterior foregut endoderm segregation defect and loss of *Nkx2.1* expression (Fig. 1c, d)[4,5]. In contrast to *Nkx2.1null* embryos, *ShhCre, Ctnnb1flox/flox* embryos did not express *Tbx4*. To eliminate the possibility that the lack of LPM caused no *Tbx4* expression phenotype, we assessed the expression of *Foxf1*, a pan-LPM marker at this developmental stage (Supplementary Fig. 2a). *Foxf1* was still expressed in the mesoderm. This observation suggests that the activation of endodermal Wnt signaling, but not *Nkx2.1* expression, is required for following mesodermal *Tbx4* expression. Thus, the initial induction of tracheal mesoderm is independent of known *Nkx2.1*-mediated respiratory endoderm development, but dependent on the activation of Wnt signaling at the ventral anterior foregut endoderm.

**Endodermal-to-mesodermal Wnt signaling induces *Tbx4* expression in tracheal mesoderm.** To further study the spatiotemporal regulation of canonical Wnt signaling during trachea–esophageal segregation at E9.5 to E11.5, we used a reporter line *LEF1EGFP* and examined the distribution of EGFP in the canonical Wnt signaling response (Fig. 2a, b)[15]. At E9.5, EGFP was detected in the ventral half of the anterior foregut endoderm where trachea endodermal cells appear and express *Nkx2.1* (Fig. 2a, b, arrowheads) and then decreased temporally at E10.5. After E10.5, the EGFP reporter was activated in the surrounding mesoderm and its intensity increased at E11.5 (Fig. 2a, b, arrowheads), which was similar to the patterning of *Axin2-LacZ*, another reporter line for the response of canonical Wnt signaling[16]. We further conducted RNAscope in situ hybridization (ISH) against *Axin2*, an endogenous Wnt target gene, to confirm activation of Wnt signaling in mesoderm. *Axin2* was highly expressed in surrounding mesoderm at E10.5 compared to endoderm, similar to the pattern observed in the reporter line (Fig. 2c). Because these Wnt-responsive mesodermal cells expressed *Tbx4* (Fig. 2b), we hypothesized that Wnt signaling in the early mesoderm is involved in the initiation of the tracheal mesoderm.

To validate the role of mesodermal Wnt signaling, we genetically ablated *Ctnnb1*, also known as *β-catenin*, which is a core component of canonical Wnt signaling, from embryonic mesoderm. We employed the *Dermo1-Cre* line, which targets embryonic mesoderm, including tracheal/lung mesoderm, and generated *Dermo1Cre, Ctnnb1flox/flox* mice[17–20]. In the mutant embryos, *Tbx4* expression was absent but *Foxf1* was retained in the mesoderm at E10.5 (Fig. 2d and Supplementary Fig. 2b), indicating that mesodermal canonical Wnt signaling is necessary for *Tbx4* expression. In contrast, endodermal *Nkx2.1* expression

and tracheoesophageal segregation were not affected, implying that mesodermal Wnt signaling and *Tbx4* is dispensable for endodermal development. Supporting the observation of lung buds in *Dermo1Cre, Ctnnb1flox/flox* embryos[20,21], the mutant lung buds still expressed *Tbx4* in mesoderm (Supplementary Fig. 3a). Disruption of Wnt signaling in the mesoderm eliminated *Tbx4* expression in the tracheal but still detectable in lung mesoderm, suggesting that Wnt-mediated mesodermal *Tbx4* induction is a unique system in trachea development but not lung development.

We further found that the *Dermo1Cre, Ctnnb1flox/flox* mutant exhibits tracheal cartilage agenesis. In the mutants, a periodic cartilage ring structure labeled with *Sox9* failed to develop at E16.5, and circumferential SM bundles labeled with smooth muscle actin (SMA) were also malformed (Fig. 2e, f) Therefore, mesodermal Wnt signaling is crucial for trachea mesenchymal development, particularly for tracheal cartilage development.

To determine whether *Tbx4* is a direct or indirect target of canonical Wnt signaling in respiratory mesoderm, we analyzed the presence of *Tcf/Lef*-binding sequences in the *Tbx4* lung mesenchyme element (*Tbx4−LME*)[22,23]. We identified five putative *Tcf/Lef*-binding sites in *Tbx4*-LME using Jaspar 2020 database (Fig. 2g)[24]. Intersecting these binding sites with public databases, such as UCSC browser and ENCODE, showed that these putative *Tcf/Lef*-binding sites are highly conserved among different vertebrates except for zebrafish and are localized in a region that contains epigenetic marks of distal regulatory elements H3K27Ac, E14.5 lung; H3K4me1, E15.5 lung; p300, postnatal day (PND) 0 lung and chromatin accessibility at E14.5 lung in which the transcription of *Tbx4* gene is active[10,25–27].

Next, we sought to identify a source of Wnt ligands that initiate mesodermal *Tbx4* expression. Due to the essential role of *Wnt2* at early tracheal/lung development[4], we conducted ISH for *Wnt2*. *Wnt2* is transiently expressed in the ventrolateral mesoderm of the anterior foregut at E9.5, which was obviously reduced by E10.5 when *Tbx4* was expressed (Figs. 2b and 3a). *Wnt2* is most likely not involved in *Tbx4* expression after E10.5. This observation prompted us to hypothesize that an endodermal-to-mesodermal interaction but not mesodermal autonomous induction is required for *Tbx4* expression. To test this hypothesis, we generated *ShhCre, Wlsflox/flox* mice, in which endodermal Wnt ligand secretion is inhibited by targeting *Wntless (Wls)* gene essential for exocytosis of Wnt ligands[28]. This endoderm-specific deletion of *Wls* resulted in loss of *Tbx4* expression in the mesoderm, but retained *Nkx2.1* expression in the endoderm and *Wnt2*, and *Foxf1* in the mesoderm (Fig. 3b, c and Supplementary Fig. 2c)[28], making these mice a phenocopy to *Dermo1Cre, Ctnnb1flox/flox* mice (Fig. 2d). *ShhCre, Wlsflox/flox* embryos also formed lung buds and expressed *Tbx4* in the distal lung mesoderm (Supplementary Fig. 3b), supporting our idea that Wnt signaling in mesoderm mainly contributes to initiation of mesodermal development of the trachea, but not of the lung. These findings indicate that the endodermal Wnt ligands are sufficient for trachea mesodermal development. From these observations, we conclude that mesodermal *Wnt2* activates endodermal canonical Wnt signaling to promote endodermal Wnt ligand expression independent of *Nkx2.1*. These Wnt ligands then induce endodermal-to-mesodermal canonical Wnt signaling to initiate tracheal mesoderm specific identity (Fig. 3d). These results also suggest that specification in the trachea endodermal lineage is not necessary for the initial induction of the tracheal mesoderm.

In the developing mouse trachea, several Wnt ligands are expressed in the endoderm between E11.5 to E13.5, such as *Wnt3a, 4, 5a, 6, 7b, 11*, and *16*[21,28]. Current single-cell RNA-seq data have shown the presence of several Wnt ligands including *Wnt4, 5a, 5b, 6*, and *7b* in the respiratory endoderm of mouse

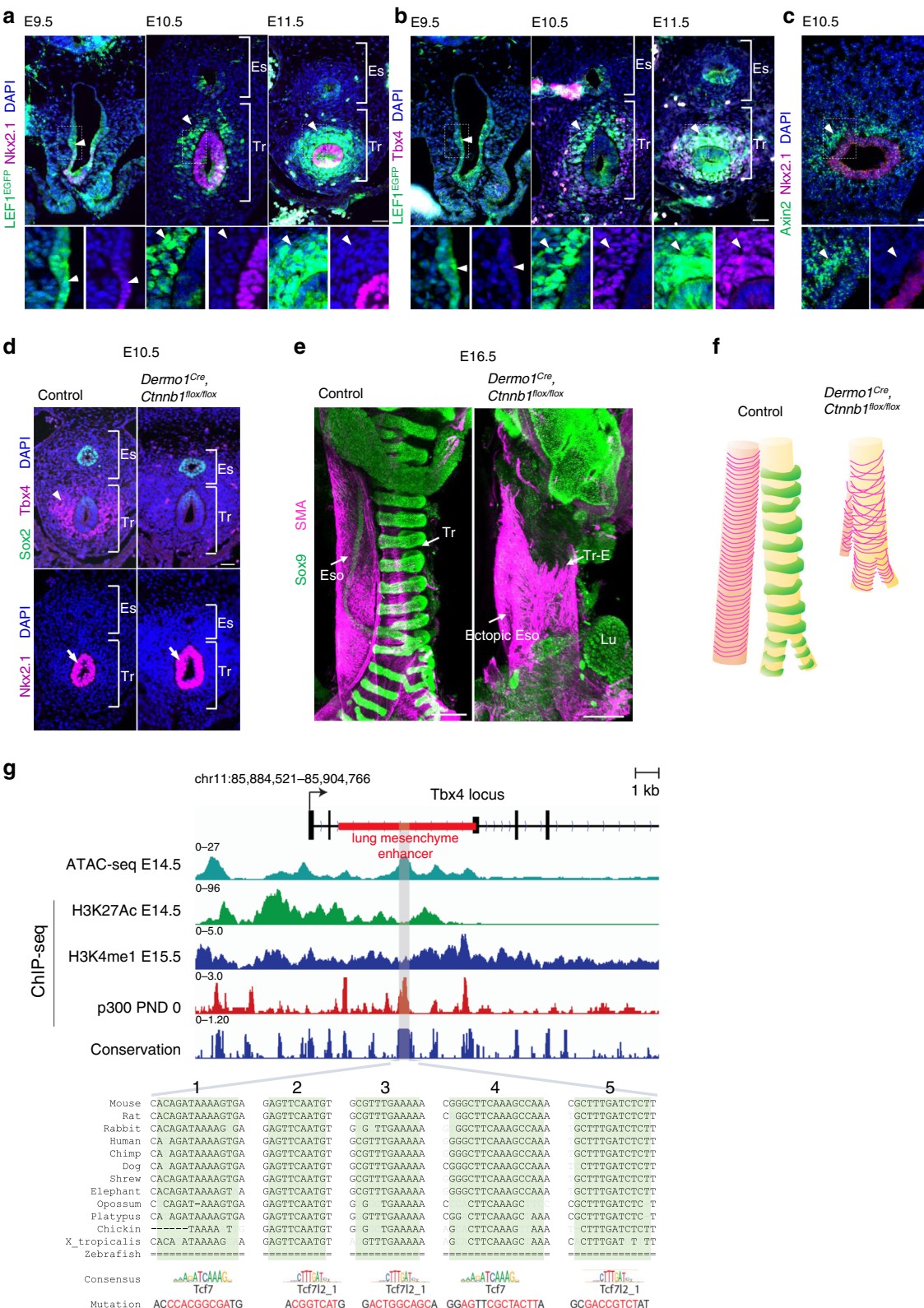

embryos at E9.5[29]. We performed ISH against these Wnt ligands at E10.5 to determine the particular ligand inducing *Tbx4* expression in trachea development (Fig. 3e, Supplementary Fig. 5). *Wnt4* was expressed in esophageal mesoderm and barely detected in tracheal endoderm. *Wnt5a, 5b,* and *6* were detected in both the endoderm and mesoderm of the trachea. More importantly, *Wnt7b* was abundantly expressed in tracheal endoderm, suggesting that *Wnt7b* might be responsible for the ensuing induction of mesodermal *Tbx4* expression.

**In vitro recapitulation of trachea mesodermal development using mouse and human embryonic stem cells.** By recapitulating developmental processes in vitro, trachea/lung endodermal

**Fig. 2 Wnt signaling is activated to promote mesodermal development of the mouse trachea. a** Transverse sections of *LEF1*^EGFP reporter mouse embryos at E9.5 to E11.5. Sections were stained for EGFP (*green*), Nkx2.1 (*magenta*), and DAPI (*blue*). Arrowheads indicate GFP+ cells. $n = 3/3$ embryos. **b** Transverse sections of *LEF1*^EGFP reporter mouse embryos at E9.5 to E11.5. Sections were stained for EGFP (*green*), Tbx4 (*magenta*), and DAPI (*blue*). Arrowheads indicate GFP+ cells. $n = 3/3$ embryos per genotype. **c** Transversal section of mouse embryo at E10.5. Section were stained for Axin2 (*green*), Nkx2.1 (*magenta*) and DAPI (*blue*) by RNAscope experiment. Arrowheads indicate Axin2+ cells. $n = 2/2$ embryos. **d** Transverse sections of *Dermo1*^Cre, *Ctnnb1*^flox/flox mouse embryos and littermate controls at E10.5. Upper panels show sections stained for Sox2 (*green*), Tbx4 (*magenta*), and DAPI (*blue*). Lower panels show sections stained for Nkx2.1 (*magenta*) and DAPI (*blue*). Arrowhead indicates Tbx4+ cells. Arrows indicate Nkx2.1+ cells. $n = 3/3$ embryos per genotype. **e** Three-dimensional imaging of whole trachea and esophagus tissue at E16.5. Cartilage morphology and smooth muscle architecture in the tracheas of *Dermo1*^Cre, *Ctnnb1*^flox/flox mouse embryos and control littermates. Whole trachea and esophagus were stained for Sox9 (*green*) and SMA (*magenta*). $n = 3/3$ embryos per genotype. **f** Model of tracheal architecture in *Dermo1*^Cre, *Ctnnb1*^flox/flox mouse embryos and control littermates based on **e**. **g** Integrative Genomics Viewer (IGV) snapshot of mm10 (chr11:85,884,521-85,904,766) showing mouse *tbx4* lung mesenchyme specific element (LME) and compiled ENCODE data of ATAC-seq E14.5 lung (ENCSR335VJW), H3K27Ac E14.5 lung (ENCSR452WYC), H3K4me1 E15.5 lung (ENCFF283EBS), EP300 postnatal day (PND) 0 lung and vertebrate conservation (Phastcons). Numbers indicate fold enrichment over input (ChIP-seq). CisBP and Jaspar predicted Tcf/Lef-binding sites (highlighted in green, region: mm10, chr11:85,893,703-85,894,206) are localized at the ATAC-seq and p300 peaks that are conserved among most vertebrates. Sequence in red shows the Tcf/Lef-binding sites mutated. Eso Esophagus, Lu Lung, Tr Trachea, Tr-E Tracheoesophageal tube. Scale bar: 40 μm (**a**, **b**), 50 μm (**c**, **d**), 300 μm (**e**).

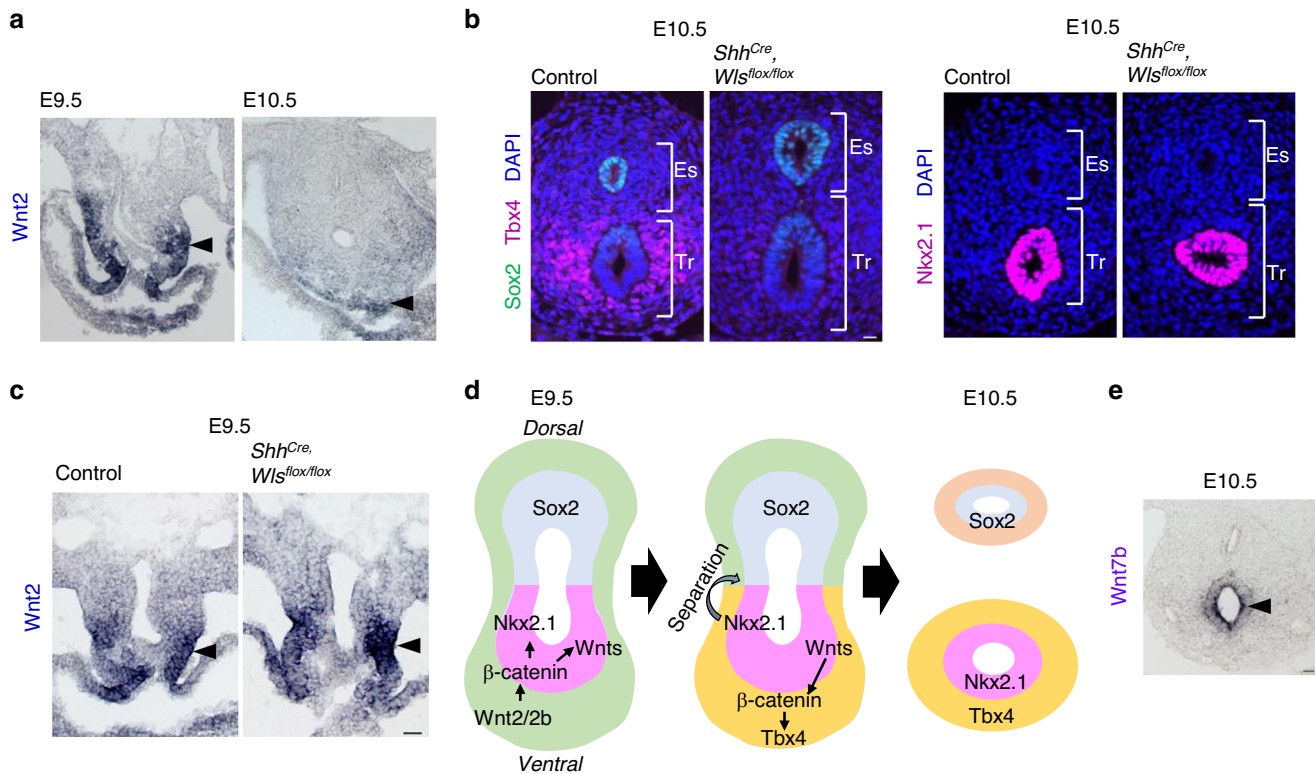

**Fig. 3 Endodermal Wnt ligands induce *Tbx4* expression for tracheal mesoderm development of mouse trachea. a** In situ hybridization for *Wnt2* mRNA during tracheoesophageal segregation. Arrowheads indicate *Wnt2* expression in the ventrolateral mesoderm at E9.5 and E10.5. $n = 2/2$ embryos. **b** Transverse sections of *Shh*^Cre, *Wls*^flox/flox mouse embryos and littermate controls at E10.5. Left panels show sections stained with Sox2 (*green*), Tbx4 (*magenta*), and DAPI (*blue*). Right panels show sections stained for Nkx2.1 (*magenta*) and DAPI (*blue*). $n = 3/3$ embryos per genotype. **c** In situ hybridization for Wnt2 mRNA in *Shh*^Cre, *Wls*^flox/flox mouse embryos and littermate controls at E9.5. Arrowheads indicate Wnt2 expression in the ventrolateral mesoderm. $n = 2/2$ embryos. **d** Refined model of tracheoesophageal segregation and tracheal mesodermal differentiation. **e** In situ hybridization for Wnt7b mRNA in mouse embryo at E10.5. Arrowhead indicates Wnt7b+ cells. $n = 2/2$ embryos. Eso Esophagus, Lu Lung, Tr Trachea. Scale bar; 40 μm (**a-c**), 50 μm (**e**).

cells and differentiated epithelial populations have been generated from both mouse and human pluripotent stem cells[30–32], and can also be used for disease modeling[33–35]. However, an established protocol for inducing tracheal/lung mesoderm and differentiated mesenchymal tissue from pluripotent cells has not yet been reported because signaling pathways coordinating the mesodermal development are still undefined. To examine whether Wnt signaling is capable of initiating the differentiation of naïve mesodermal cells to Tbx4+ trachea mesodermal cells in vitro, we established a protocol for LPM induction from mouse ESCs (mESCs) by refining the published protocol for LPM induction from human pluripotent stem cells[36]. Because mouse and human ESCs show different states called naïve and primed, which correspond to pre and postimplantation epiblasts, respectively, we converted mESCs into an epiblast 'primed' state that led to middle-primitive streak (mid-PS) cells[37]. These mid-PS cells were then differentiated into LPM cells (Fig. 4a). At day 5, LPM induction was confirmed by immunocytochemistry (ICC) for

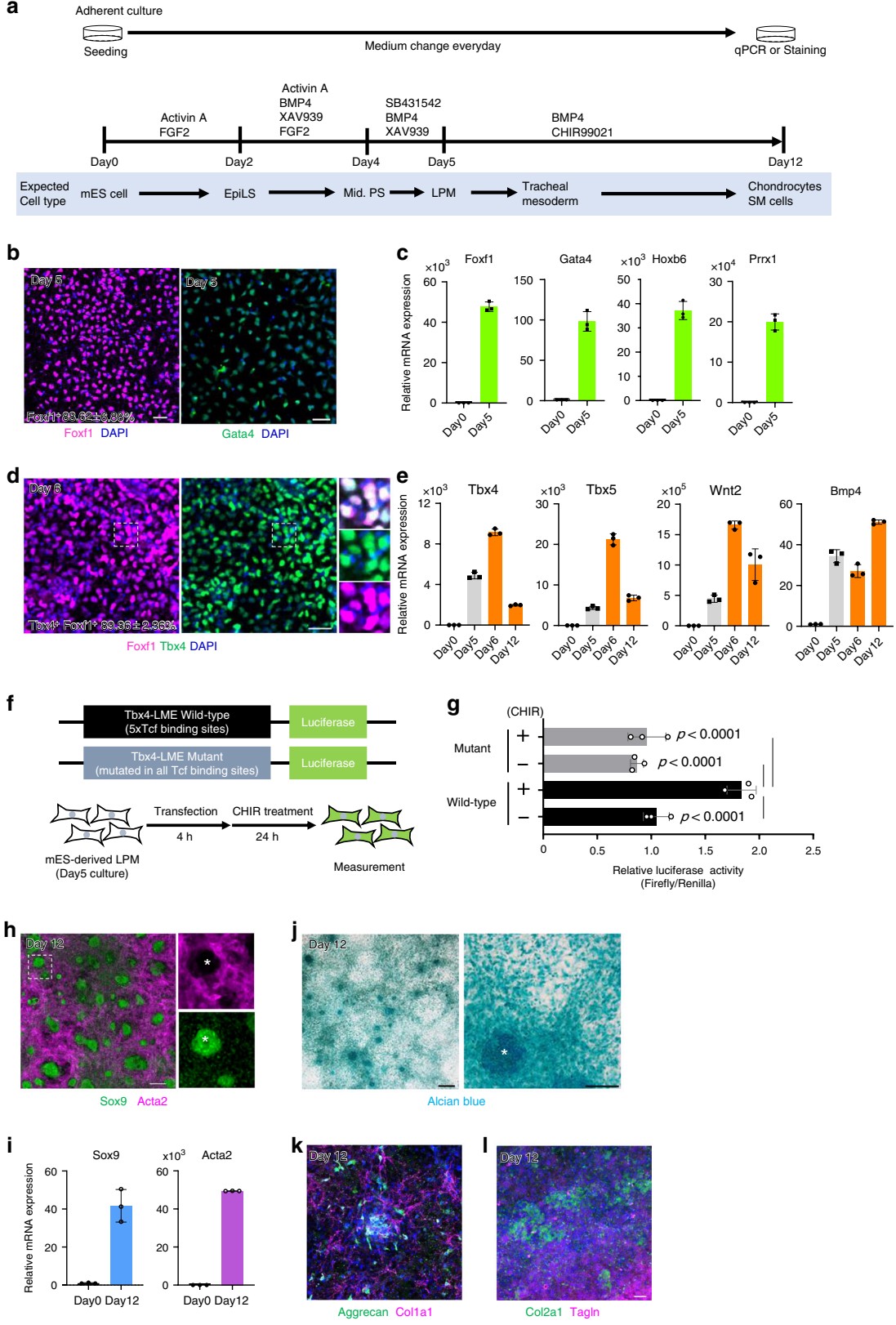

*Foxf1* and *Gata4*, which are known to be expressed in LPM including splanchnic mesoderm[38,39] (Fig. 4b). ICC showed that 89% of total cells were Foxf1⁺ LPM. Furthermore, qRT-PCR also showed obvious upregulation of LPM marker genes, such as *Foxf1*, *Gata4*, *Hoxb6*, *Prrx1*, and *Bmp4* (Fig. 4c and e)[40–44]. Given that previous mouse genetic studies have identified *Bmp4* as a

crucial regulator of trachea development[6,45], we tested whether canonical Wnt and Bmp4 signaling are sufficient to direct the differentiation of LPM into the tracheal mesoderm (Foxf1⁺/Tbx4⁺). mESC-derived LPM cells were cultured with CHIR99021, a GSK3β inhibitor to stabilize β-catenin and activate canonical Wnt signaling, and *Bmp4*. At day 6, 89% of total cells

**Fig. 4 Generation of trachea mesodermal cells and chondrocytes from mouse ESCs in vitro. a** Experimental design to generate tracheal mesoderm from mESCs. **b** Differentiating cells from mESCs at day 5. Cells were stained for Foxf1 (*magenta*) and Gata4 (*green*), respectively. % was calculated from randomly chosen 3 fields. Images are representative of two independent experiments. **c** qRT-PCR for LPM markers of mESC-derived LPM ($n = 3$ independent wells). Experiments were repeated at least twice. **d** Differentiating cells from mESCs at day 6. Cells were stained for Tbx4 (*green*) and Foxf1 (*magenta*). % was calculated from randomly chosen three fields. Images are representative of at least two independent experiments. **e** qRT-PCR for respiratory mesoderm marker expression of mESC-derived trachea mesodermal cells. ($n = 3$ independent wells). Experiments were repeated at least twice. **f** Diagram showing the constructs utilized in luciferase experiments containing *Tbx4*-−−LME Tbx4-LME wild-type containing five Tcf/Lef-binding sites and Mutant. mESC-derived LPMs were transfected with wild-type or mutant *Tbx4*-LME during 4hrs following by respiratory induction in presence or absence of CHIR99021. **g** Luciferase assay examining the activation of *Tbx4*-LME wt and mutant in response to 3 μM CHIR99021. *P*-values were provided by two-sided Tukey's multiple comparison. ***$p < 0.0001$ ($n = 3$ from independent wells from a single experiment). **h** Differentiating cells from mESCs at day 12. Cells were stained for Acta2 (*magenta*) and Sox9 (*green*). The asterisk indicates Sox9$^+$/SMA$^-$ chondrocyte aggregates. Images are representative of two independent experiments. **i** qRT-PCR for *Sox9* and *Acta2* expression of hESC-derived trachea mesodermal cells ($n = 3$ independent wells). Experiments were repeated twice. **j** Differentiating cells from mESCs at day 12. Chondrocytes were stained with Alcian blue. The asterisk indicates one of the chondrocyte aggregates. Images are representative of two independent experiments. **k** Differentiating cells from mESCs at day 12. Cells were stained for Col1a1 (*magenta*) and Aggrecan (*green*). Image is representative of two independent experiments. **l** Differentiating cells from mESCs at day 12. Cells were stained for Tagln (*magenta*) and Col2a1 (*green*). Image is representative of two independent experiments. Each column shows the mean with S.D. Scale bar; 50 μm. Source data for **b, c, d, e, g, i** are provided in Source data file.

became double positive for Foxf1 and Tbx4. qRT-PCR further demonstrated elevated expression of tracheal marker genes such as *Tbx5, Wnt2, Bmp4* in addition to *Tbx4* (Fig. 4d, e). To further confirm the respiratory characteristics of these cells, we took advantage of the 5 *Tcf/lef*-binding sequences in *Tbx4*-LME, which we described in Fig. 2g. We established a luciferase reporter assay by reporter plasmids that express luciferase under the control of *Tbx4*-LME (Fig. 4f). The reporter plasmid was transfected into mESC-derived LPM and luciferase activity was assessed during differentiation. After 24 h (at day 6), the luciferase activity significantly increased in the presence of CHIR99021 (Fig. 4g). Importantly, the mutated reporter, in which all *Tcf/Lef*-binding sequences were changed to random sequences (Figs. 2g and 4f), did not respond to CHIR99021. The modest increase of luciferase activity might be due to low transfection efficiency. These results determined that the mESC-derived cells were differentiated into proper tracheal mesoderm at day 6.

Because tracheal mesenchyme includes cartilage and smooth muscle, we wondered whether our protocol induces mESC to differentiate into these tissues. At day 12, Sox9$^+$ aggregated cell masses positive for Alcian blue staining appeared on the dish, indicative of chondrocytes (Fig. 4h–j). Smooth muscle cells (SMA$^+$ cells) concurrently appeared to show fibroblastic morphology and filled the space not filled by the Sox9$^+$ cells (Fig. 4h, i). Other chondrogenic markers (*Aggrecan, Collagen2a1, Sox5/6, Epiphycan*) and smooth muscle markers (*Tagln, Collagen1a1*) were also present in the differentiated cells (Fig. 4k, l and Supplementary Fig. 7a). These data suggest that the mESC-derived tracheal mesoderm is able to develop into tracheal mesenchyme, including chondrocytes and smooth muscle cells.

Finally, we tested the role of Wnt signaling in the human tracheal mesoderm using human ESCs (hESCs). Human LPM induction was performed by following an established protocol[36] (Fig. 5a). Subsequently, the cells were directed to tracheal mesoderm by using CHIR99021 and BMP4. For validating hESC-derived LPM, we checked the common LPM markers at day 2 and confirmed that these markers were abundantly expressed in the LPM (Fig. 5b, c and Supplementary Fig. 8k). Immunostaining determined that 95% of the total cells expressed *FOXF1* at day 2 (Fig. 5b). Because *Tbx4* is also expressed in the limbs and other fetal mouse tissues[23], we sought additional genetic markers for the tracheal mesoderm. We searched the single-cell transcriptomics dataset of the developing splanchnic mesoderm at E9.5 and identified *Nkx6.1* as a marker for mesodermal cells surrounding the trachea, lung, and esophagus[29].

We performed immunostaining and found that *Nkx6.1* was expressed in tracheal and esophageal mesenchyme throughout development (Supplementary Fig. 6). Of note, *Nkx6.1* was expressed in esophageal and dorsal tracheal mesenchyme but not ventral trachea, which enabled us to define three subtypes of tracheal-esophageal mesenchyme based on the combination of *Tbx4* and *Nkx6.1* expression (i.e., Tbx4$^+$/Nkx6.1$^+$; dorsal tracheal mesenchyme, Tbx4$^+$/Nkx6.1$^-$; ventral tracheal mesenchyme, Tbx4$^-$/Nkx6.1$^+$; esophageal mesenchyme) (Supplementary Fig. 6). Having characterized the subtypes of tracheoesophageal mesoderm in vivo, the expression of *TBX4* and *NKX6.1* in the hESC-derived tracheal mesoderm was examined by ICC and qRT-PCR. Although *TBX4* was induced in a Wnt activator dose-dependent manner, *NKX6.1* expression was not significantly elevated (Supplementary Fig. 8a, b), suggesting that human trachea mesodermal development requires an additional factor to become more in-vivo-like. Because the ventral LPM is exposed to *SHH* in addition to *Wnt* and *Bmp4* during tracheoesophageal segregation[46,47], we assessed whether the SHH activator (PMA; purmorphamine) can improve differentiation from hESC-derived LPM cells into the tracheal mesoderm. As expected, both *TBX4* and *NKX6.1* expression was upregulated by the SHH activator (Supplementary Fig. 8c, d). After day 5, the differentiating cells also expressed respiratory markers such as *TBX4, TBX5, WNT2, BMP4*, and *NKX6.1* (Fig. 5d–h, Supplementary Fig. 8k). In this culture condition, CHIR99021 enhanced the expression of *TBX4* and *NKX6.1* genes in a dose-dependent manner (Supplementary Fig. 8e, f). We also performed qPCR analysis for *FOXF1* expression as a pan-LPM marker. *FOXF1* expression was decreased by CHIR99021 but still retained after induction (Supplementary Fig. 8g). This result also reflects the feature of in vivo tracheal mesoderm because *FOXF1* expression is decreased in the ventral mesoderm of the mouse trachea (Supplementary Fig. 2a–c). These data indicate that HH/BMP was sufficient to induce *FOXF1* expression but HH/BMP/WNT was necessary for human respiratory mesoderm induction. We further estimated the efficiency of the induction by immunostaining for *TBX4* and *FOXF1*, and then confirmed that 83% of total cells were TBX4$^+$/FOXF1$^+$ double positive cells at day 5 (Fig. 5d). At day 10, *NKX6.1* expression was clearly elevated, and 30.3% of the total cells became TBX4$^+$/NKX6.1$^+$ double positive (Fig. 5e–g) while 18.0% were TBX4$^+$/NKX6.1$^-$ (Fig. 5g). These data suggest that the half of the cells induced with our protocol are trachea mesodermal cells. Further extended culture induced SOX9$^+$ aggregates, which were positive for Alcian blue staining in

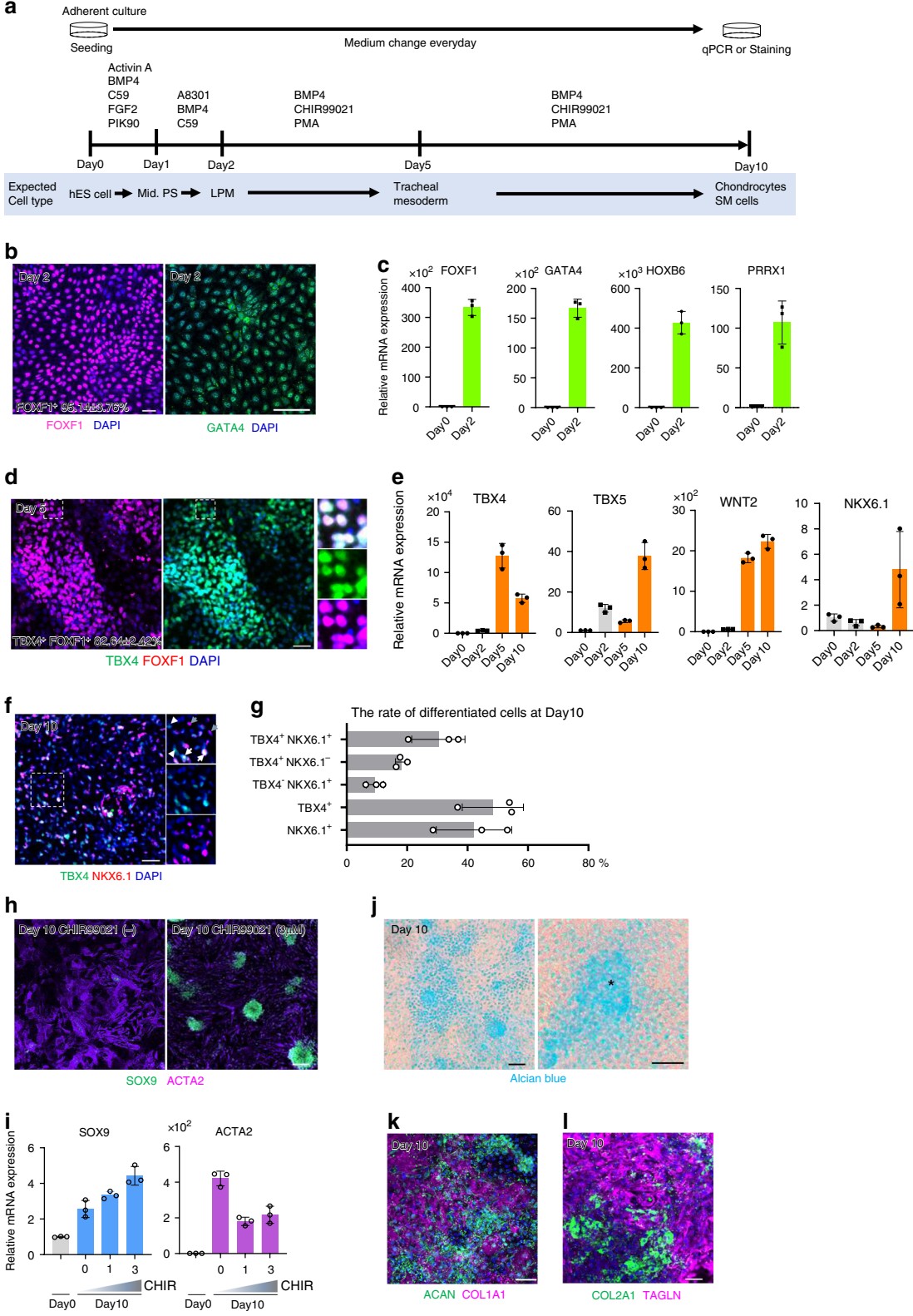

a Wnt activity-dependent manner (Fig. 5h–l). Likewise in mESC-derived cells, ACTA2$^+$ smooth muscle-like fibroblastic cells occupied Sox9$^-$ region (Fig. 5i). These cells also expressed chondrogenic markers and smooth muscle cell markers (Fig. 5k, l and Supplementary Fig. 7b).

In this culture system, the removal of *BMP4* from the growth factor cocktail did not affect differentiation, implying that exogenous *BMP4* activation is dispensable (Supplementary Fig. 8h–j). Because of the obvious upregulation of the endogenous *BMP4* gene in the hESC-derived LPM by day 2, endogenous *BMP4* may be enough to induce tracheal mesoderm and chondrocytes (Supplementary Fig. 8k). Taken together, these data suggest that Wnt signaling plays a unique role in driving differentiation into tracheal mesoderm and

**Fig. 5 Generation of trachea mesodermal cells and chondrocytes from human ESCs in vitro. a** Experimental design to generate tracheal mesoderm from hESCs. **b** Differentiating cells from hESCs at day 2. Cells were stained for FOXF1 (*magenta*) and GATA4 (*green*), respectively. % was calculated from randomly chosen 3 fields. Images are representative of two experiments. **c** qRT-PCR for LPM marker expression of hESC-derived LPM ($n = 3$ independent wells). Experiments were repeated at least twice. **d** Differentiating cells from hESCs at day 5. Cells were stained for TBX4 (*green*) and FOXF1 (*magenta*). % was calculated from randomly chosen three fields. Images are representative of three wells in a single experiment. **e** qRT-PCR for respiratory mesoderm marker expression of hESC-derived trachea mesodermal cells ($n = 3$ independent wells). Experiments were repeated at least twice. **f** Differentiating cells from hESCs at day 10. Cells were stained for TBX4 (*green*) and NKX6.1 (*magenta*). Images are representative of three wells in a single experiment. White arrows; TBX4$^+$/NKX6.1$^+$ mesodermal cells. White arrowheads; TBX4$^+$/NKX6.1$^-$ mesodermal cells. Grey arrows; TBX4$^-$/NKX6.1$^+$ mesodermal cells. **g** The rate of differentiated cells at day 10. % was calculated from randomly chosen three fields. Images are representative of three wells in a single experiment. **h** Differentiating cells from hESCs with or without CHIR99021 at day 10. Cells were stained for ACTA2 (*magenta*) and SOX9 (*green*). Images are representative of at least two experiments. **i** qRT-PCR for *SOX9* and *ACTA2* expression of hESC-derived trachea mesodermal cells with different doses of CHIR99021 ($n = 3$ independent well). Experiments were repeated twice. **j** Differentiating cells from hESCs at day 10. Chondrocytes were stained with Alcian blue. The asterisk indicates a chondrocyte aggregate. Images are representative of two experiments. **k** Differentiating cells from hESCs at day 10. Cells were stained for COL1A1 (*magenta*) and AGGRECAN (*green*). Images are representative of two experiments. **l** Differentiating cells from hESCs at day 10. Cells were stained for TAGLN (*magenta*) and COL2A1 (*green*). Images are representative of two experiments. Each column shows the mean with S.D. ($n = 3$). Scale bar; 50 μm (**d, f, h, j, k, l**), 100 μm (**b**). Source data for **b, c, d, e, g, i** are provided in Source data file.

chondrocytes from the LPM, which is conserved between mice and human.

## Discussion

This study demonstrates that endodermal-to-mesodermal canonical Wnt signaling is the cue that initiates trachea mesodermal development in developing mouse embryos, which is independent of the previously known *Nkx2.1*-mediated respiratory tissue development. Based on our knowledge of developmental biology, we successfully generated tracheal mesoderm and chondrocytes from mouse and human ESCs. In our protocol, we stimulated ESC-derived LPM with Wnt, Bmp and SHH signaling to mimic spatial information of the ventral anterior foregut. For induction of respiratory endoderm, Wnt, Bmp, and Fgf signaling are required to direct cells in anterior foregut to differentiate into the respiratory lineage[30–32]. Thus, Wnt, and Bmp signaling are conserved factors that provide spatial information, while Fgf and SHH are required in endoderm and mesoderm induction, respectively, reflecting the unique signaling pathways in each tissue. Mesoderm induction may need fewer exogenous growth factors because the mesodermal cells themselves are sources of spatial information, such as *BMP4* in our protocol.

In our culture system, *Bmp4* and Wnt activator were sufficient to induce tracheal mesoderm from mouse ES-derived LPM, but HH signalling was also required for the induction of human tracheal mesoderm. To investigate the involvement of HH signalling in mesodermal development, we examined *Shh-null* mouse embryo (Supplementary Fig. 9). Reflecting the observation in in vitro differentiation, *Tbx4* was still expressed in mutant, suggesting the dispensable role of HH signalling in mouse tracheal mesoderm. By contrast, in human, HH signalling is involved in TEF, but the characteristics of mesoderm is not defined yet[48]. In future study, it would be important to decipher the role of HH signalling on human trachea mesodermal development.

In chicken embryo, misexpression of *Tbx4* throughout the respiratory-esophageal region ectopically induces *Nkx2.1* expression in distal esophageal endoderm underneath manipulated *Tbx4*-expressing mesoderm[49], indicating that mesodermal *Tbx4* expression defines tracheal identity in chicken. However, in mouse, we determined that *Nkx2.1* expression was not affected in Wnt mutant (e.g., *Dermo1^Cre; Ctnnb1^flox/flox*, and *Shh^Cre, Wls^flox/flox*) although *Tbx4* expression was completely lost (Figs. 2d and 3b), suggesting that mesodermal *Tbx4* expression does not trigger tracheal endoderm specification, which is consistent with retaining *Nkx2.1* expression in respiratory explants from *Tbx4*-

null mouse embryos[10]. These findings indicate that the mechanism inducing tracheal identity is different between mouse and chicken.

Another question is how trachea and lung are differentially specified, despite the strong commonalities between them. Canonical Wnt signalling induces *Tbx4* expression in the tracheal mesoderm but not in the lungs. The trachea and lungs share the maker genes, such as endodermal *Nkx2.1* and mesodermal *Tbx4*, but recent mouse genetics have accumulated the evidence that different combination of geneset regulated the trachea and the lungs individually. For instance, *FGF10-null* mouse embryos failed lung development, but still develop the trachea[50]. In contrast, *Bmpr1a, b-null* embryos show the tracheal agenesis phenotype while the mutant embryos have the lung buds[6]. It would be interesting to examine expression patterns of receptors for *Fgf10, Bmp*, and *Wnt* in the trachea and lung that may distinguish these organs.

Past studies have proposed that Wnt signaling is required for cartilage formation in the late stage of trachea development[19,28]. Tracheal chondrocytes originate in *Tbx4*-expressing cells, and *Tbx4/5* knockout mice show severe defects in chondrogenesis[10,22]. The early loss of *Tbx4* expression in Wnt mutant might contribute, at least a part, the abnormal structures of cartilage in the late stage of development.

In this study, we were unable to perform tissue-specific targeting for trachea endodermal or mesodermal cells because of multiple Cre-expression patterns in *Shh-Cre* and *Dermo1-Cre* mouse lines. For example, *Shh* is also expressed in the notochord and ventral neural tube[51]. Future studies with analyses of respiratory tissue-specific Cre lines would strengthen the evidence demonstrating that mutual interaction between respiratory endoderm and mesoderm is required for the induction of trachea development.

*Dermo1^Cre, Ctnnb1^flox/flox* mutants display a tracheal cartilage agenesis phenotype. Due to the multiple functions of *Ctnnb1* in transcriptional regulation and cellular adhesion, however, it is possible that *Ctnnb1* knockout affects not only Wnt-mediated transcriptional regulation but also mesenchymal cell-cell adhesion[19]. To exclude this possibility, we examined the distribution of *Cdh2* as an adhesive molecule in tracheal mesoderm. *Cdh2* expressions in the ventral half of tracheal mesoderm were indistinguishable between control and *Dermo1^Cre, Ctnnb1^flox/flox* embryos (Supplementary Fig. 3). Furthermore, our luciferase assay showed that respiratory mesenchyme specific *cis*-regulatory region of *Tbx4* is stimulated by CHIR99021 through Tcf/Lef-binding elements in the developing tracheal mesoderm in vitro.

These findings suggest that Wnt signaling-mediated transcriptional regulation is important for the induction of tracheal mesoderm.

Recently, Han et al. delineated mesodermal development during organ bud specification using single-cell transcriptomics analyses of mouse embryos from E8.5 to E9.5[29]. Based on the trajectory of cell fates and signal activation, this group also generated organ-specific mesoderm, including respiratory mesoderm, from hESCs, thereby determining that *Wnt, BMP4, SHH*, and retinoic acid direct differentiation of hESC-derived splanchnic mesoderm to respiratory mesoderm, supporting our current findings. In our protocol, retinoic acid was not included in the media, but alternatively vitamin A, a precursor for retinoic acid, was supplemented. Endogenous metabolite of vitamin A might affect the differentiation into respiratory mesoderm.

These culture methods could be a strong tool to study human organogenesis and the aetiology of TEA and TA, as well as to provide cellular resource for human tracheal tissue repair.

## Methods

**Mice**. All mouse experiments were approved by the Institutional Animal Care and Use Committee of RIKEN Kobe Branch. Mice were handled in accordance with the ethics guidelines of the institute. Mice were housed in 18–23 °C with 40–60% humidity. A 12-h light/12-h dark cycle was used. *Nkx2.1[null], Shh[Cre], Dermo1[Cre], Ctnnb1[flox/flox], Wls[flox/flox]* mice were previously generated[13,18,52–54].

In all experiments, at least three embryos from more than two littermates were analyzed. All attempts for replicate were successful. Sample size was not estimated by statistical methods. No data were excluded in this study. All control and mutant embryos were analyzed. We did not distinguish the sex of the embryos. No blinding was done in this study.

**Immunostaining**. Mouse embryos were fixed by 4% Paraformaldehyde/PBS (PFA) at 4 °C overnight. Specimens were dehydrated by ethanol gradient and embedded in paraffin. Paraffin sections (6-μm) were deparaffinized and rehydrated for staining. Detailed procedure and antibodies of each staining were listed in Supplementary Table 1.

**In situ hybridization**. Mouse embryos were fixed with 4% PFA/PBS at 4 °C overnight, and then tracheas were dissected. Specimens were incubated in sucrose gradient (10, 20, and 30%) and embedded in OCT compound. Frozen sections (12-μm) were subjected to in situ hybridization. For Wnt2, 4, 5a, 7b probe construction, cDNA fragments were amplified by primers listed in Supplementary Table 2. These cDNA fragments were subcloned into pBluscript SK+ at *Eco*RI and *Sal*I sites. For Wnt5b and six probes, pSPROT1-Wnt5b (MCH085322) and pSPROT1-Wnt6 (MCH000524) were linearized at *Sal*I sites, The NIA/NIH Mouse 15 K and 7.4 K cDNA Clones were provided by the RIKEN BRC[55–57]. Antisense cRNA transcripts were synthesized with DIG labeling mix (Roche Life Science) and T3 or SP6 RNA polymerase (New England Biolabs Inc.). Slides were permeabilized in 0.1% Triton-X100/PBS for 30 min and blocked in acetylation buffer. After prehybridization, slides were hybridized with 500 ng/ml of DIG-labeled cRNA probes overnight at 65 °C. After washing with SSC, slides were incubated with anti-DIG-AP antibodies (1:1000, Roche Life Science, 11093274910). Sections were colored with BM-purple (Roche Life Science, 11442074001).

For RNAscope experiments, the RNAscope Multiplex Fluorescent v2 assays (Advanced Cell Diagnostics, 323110) were used. The detailed procedure and probes were listed on Supplementary Table 3.

**Cell culture**. For mesodermal differentiation from mES cells, C57BL/6J-Chr 12 A/J/NaJ AC464/GrsJ mES cells (The Jackson Laboratory) and EB3 cells (AES0139, RIKEN BioResource Center) were used. C57BL/6J-Chr 12 A/J/NaJ AC464/GrsJ mES cells were kindly provided by Kentaro Iwasawa and Takanori Takebe (Center for Stem Cell & Organoid Medicine (CuSTOM), Perinatal Institute, Division of Gastroenterology, Hepatology and Nutrition, Cincinnati Children's Hospital, Cincinnati). EB3 was kindly provided by Dr. Hitoshi Niwa (Department of Pluripotent Stem Cell Biology, Institute of Molecular Embryology and Genetics in Kumamoto University)[58,59]. Cells were maintained in 2i + leukemia inhibitory factor (LIF) media (1000 units ml⁻¹ LIF, 0.4 μM PD0325901, 3 μM CHIR99021 in N2B27 medium)[37] on ornithine-laminin coated-dishes[37]. For mesodermal differentiation of mouse ES cells, cells were digested by TrypLE express (Thermo Fisher Scientific, 12604013) and seeded onto Matrigel-coated 12-well plate. EpiLC were induced by EpiLC differentiation medium (1% knockout serum, 20 ng ml⁻¹ Activin A, 12 ng ml⁻¹ FGF2, and 10 μM Y27632 in N2B27 Medium)[37] for 2 days. Lateral plate mesoderm was established by Loh's protocol with some modification[36]. EpiLC cells were digested by TrypLE express to single cells and seeded onto Matrigel-coated 12-well plate at the density of $6 \times 10^5$ cells per well. The cells around middle-

primitive streak was induced by LPM D2 medium composed of 2% B27 Supplement Serum free (Thermo Fisher Scientific, 17504044), 1 × GlutaMax (Thermo Fisher Scientific, 35050061), 20 ng ml⁻¹ basic FGF (Peprotech, AF-100-18B), 6 μM CHIR99021 (Sigma–Aldrich, SML1046), 40 ng ml⁻¹ BMP4 (R&D Systems, 5020-BP-010), 10 ng ml⁻¹ Activin A (Peprotech, PEP-120-14-10), 10 μM Y27632 (Sigma–Aldrich, Y0503) in Advanced DMEM (Thermo Fisher Scientific, 12491015) for 48 h. After that, LPM was induced by LPM D4 medium composed of 2% B27 Supplement Serum free, 1 x GlutaMax, 2 μM XAV939 (Sigma–Aldrich, X3004), 2 μM SB431542 (Merck, 616461), 30 ng ml⁻¹ human recombinant BMP4 in Advanced DMEM for 24 h. At Day 5, respiratory mesenchyme was induced by Day 5 medium composed of 2% B27 Supplement Serum free, 1 x GlutaMax, 1 μM CHIR99021, 10 ng ml⁻¹ BMP4. Medium were freshly renewed everyday.

H1 (NIHhESC-10-0043 and NIHhESC-10-0062), human embryonic stem cell, was provided by Cincinnati children's hospital medical center Pluripotent Stem Cell Facility. Cells were maintained in mTeSR1 medium (Stem Cell Technologies) on Matrigel-coated plate. For differentiation of H1 cells to mesodermal cells, confluent cells were digested by Accutase to single cells and seeded onto Geltrex-coated 12-well plate at the dilution of 1:20–1:18 in mTeSR1 with 1 uM Thiazovivin (Tocris). Next day, the cells around middle-primitive streak were induced by cocktails of 6 μM CHIR99021 (Sigma–Aldrich, SML1046), 40 ng ml⁻¹ BMP4 (R&D Systems, 5020-BP-010), 30 ng ml⁻¹ Activin A (Cell Guidance Systems), 20 ng ml⁻¹ basic FGF (Thermo Fisher Scientific), and 100 nM PIK90 (EMD Millipore) in Advanced DMEM/F12 including 2% B27 Supplement minus vitamin A, 1% N2 Supplement, 10 uM Hepes, 100UI mL⁻¹ Penicillin/Streptomycin, 2 mM L-glutamine for 24 h. After that, LPM was induced by LPM D2 medium composed of 1 μM Wnt C59 (Cellagen Technologies), 1 μM A83-01 (Tocris), 30 ng ml⁻¹ human recombinant BMP4 in Advanced DMEM/F12 including 2% B27 Supplement minus V. A., 1 x N2 Supplement, 10uM Hepes, 100UI mL⁻¹ Penicillin/Streptomycin, 2 mM L-glutamine for 24 h. To generate respiratory mesenchyme, we combined 3 uM CHIR99021, 2 uM Purmorphamine (Tocris), and 10 ng ml⁻¹ Bmp4 in Advanced DMEM/F12 medium including 2% B27 Supplement Serum free, 1 x N2 Supplement, 10uM Hepes, 100UI mL⁻¹ Penicillin/Streptomycin, 2 mM L-glutamine from Day 2 to Day 10. Medium was freshly renewed everyday.

**Immunocytochemistry**. At differentiating process, cells were fixed by 4% PFA for 10 min at room temperature. For intracellular staining, cells were permeabilized by 0.2% TritonX-100/PBS for 10 min at room temperature. After blocking the cells with 5% normal donkey serum or 0.3% Triron-X100/1% bovine serum albumin, cells were incubated with primary antibodies overnight at 4 °C. Then, cells were incubated with secondary antibodies for 1 h at room temperature. Detailed procedure and antibodies of each staining were listed in Supplementary Table 4.

**Luciferase reporter assay**. The fraction of mouse *Tbx4* lung mesenchyme specific enhancer (LME) (mm10, chr11:85,893,703-85,894,206, GenScript, ID U3154EL200-3)[22] or *Tbx4*-LME containing putative Tcf/Lef sites mutated (GenScript, ID U3154EL200-6) were synthesized and cloned into pGL4.23 (luc2/minP) vector (promega).

mESC-derived LPM cells were transfected at day 5 in 150 μl of Opti-MEM (Thermo Fisher Scientific, 31985088) with 2 μl of Lipofectamine Stem (Thermo Fisher Scientific, STEM00003) and 1 μg of pGL4.23 (luc2/minP) containing a fraction of mouse *Tbx4*-LME or *Tbx4*-LME containing mutated Tcf/Lef sites. Four hours after transfection the tracheal mesenchyme was induced using day 5 medium in presence or absence of Wnt activator (3 μM CHIR99021) and cells were cultured for 24 h and then lysed and assayed using Dual-Luciferase Reporter Assay System (Promega, E1980).

**Alcian blue staining**. Cells were fixed in 4% PFA/PBS for 10 min at room temperature. After washing with PBS, cells were incubated with 3% acetic acid for 3 min and then stained with 1% alcian blue/3% acetic acid for 20 min.

**Quantitative RT-PCR**. Total mRNA was isolated by using the Nucleospin kit (TaKaRa, 740955) according to manufacturer's instruction. cDNA was synthesized by Super™ Script™ VILO cDNA synthesis kit (Thermo Fisher Scientific, 11754050). qPCR was performed by PowerUp™ SYBR™ Green Master Mix on QuantStudio 3 or 6. Primer sequences were listed on Supplementary Tables 5 and 6. Data are expressed as a fold-change and were normalized with undifferentiated cells expression.

**Statistical analyses**. Statistical analyses were performed with Excel2013 (Microsoft) or PRISM8 (GraphPad software). For multiple comparison, one-way ANOVA and two-tailed Tukey's methods were applied. For paired comparison, statistical significance was determined by F-test and Student's or Welch's two-tailed *t*-test.

**Reporting summary**. Further information on research design is available in the Nature Research Reporting Summary linked to this article.

## Data availability

The authors declare that all data supporting the findings of this study are available within the article and its Supplementary Information files or from the corresponding author upon reasonable request. Source data are provided with this paper (Figs. 4b–e, g, i, 5b–e, g, i and Supplementary Figs. 7 and 8). The datasets generated during the current studies are available in the System Science of Biological Dynamics (SSBD) database[60]. Source data are provided with this paper.

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

## Acknowledgements

We thank Hinako M Takase and Hiroshi Hamada for Wntless conditional flox mice, Masatoshi Takeichi for Ctnnb1 conditional flox mice, Hironobu Fujiwara for LEF1-EGFP mice, and Animal Resource Development Unit. Hiroshi Niwa kindly provided EB3, mouse ES cells. Kentaro Iwasawa and Takanori Takebe kindly provided C57BL/6 mouse ES cell line. We are grateful to Debora Sinner for the help on RNAscope experiment. We thank Scott Rankin for the help on the enhancer analyses of Tbx4 genes. We also thank Shunsuke Mori and Mototsugu Eiraku for Tbx4 antibody. We thank Yuka Noda and David Luedeke for general technical support. We also thank Shigeo Hayashi and Hazuki Hiraga for primary reading. These studies are supported by the funding from Grants-in-Aid for Scientific Research (B) (17H04185) (20H03693) (M.M.), Young Scientists (17K15133 and 19K16156) (K.K.), Promotion of Joint International Research (A) (18KK0423) (K.K.) of the Ministry of Education, Culture, Sports, Science and Technology, Japan, and from The Takeda Science Foundation for the Life Science (M.M.), and The Uehara Memorial Foundation (K.K.). Partially supported by grant NICHD P01HD093363 to A.M.Z.

## Author contributions

K.K. and M.M. designed the project and performed experiments with the aid of A.L.M., A.Y., C.M., and K.T.F. A.M.Z. analysed single-cell transcriptomics for definitive endoderm and splanchnic mesoderm. A.L.M. performed enhancer analyses of Tbx4 gene and supported human ES cell experiments, A.Y. supported mouse experiments. K.K., K.T.F., and C.M. performed mouse ES cell experiment. C.A. and M.H. contributed to mouse and human embryonic-stem-cell-based lateral plate mesoderm induction and differentiation experiments. K.K. and M.M. wrote the manuscript with the contribution of all authors.

## Competing interests

The authors declare no competing interests.
