## [Peer Review File · Nature Communications]

Reviewers' Comments:

Reviewer #1:

Remarks to the Author:

Development of endodermal tissues such as the tracheal epithelium critically depends on surrounding mesenchymal tissues that provide critical signals (e.g., Goss et al., 2009; Dev Cell) as well as mechanical forces (e.g., Kishimoto et al., 2018; Nature Communications). While the mesenchymal tissues that surround these various endodermal tissues were once regarded as rather similar, it is now known that they are quite different from one another: for instance, the mesenchyme surrounding the respiratory tract is unique, as it specifically expresses Tbx4 from E9.0-onwards in the mouse embryo (Arora et al., 2012; PLoS Genetics). While it has been partially investigated how tracheal epithelium progenitors are specified, it has remained relatively unknown how the surrounding tracheal mesenchyme is specified. Here, Kishimoto et al. address this important issue.

Kishimoto et al. perform both *in vivo* and *in vitro* experiments to address this question. In short, the authors find the tracheal mesenchyme fails to express Tbx4 if the ability of tracheal epithelium (Shh-Cre;Ctnnb1[fl/fl]) or tracheal mesenchyme (Dermo-Cre;Ctnnb1[fl/fl]) to respond to Wnt is impaired. The fact that a similar phenotype is seen when Ctnnb1 is deleted in either the tracheal epithelium or mesenchyme is quite striking, and suggests that bidirectional Wnt-mediated communication between the two tissues is required for tracheal mesenchyme specification. The authors further show that impairing the ability of the tracheal epithelium to secrete Wnt (Shh-Cre;Wls[fl/fl]) leads to the same phenotype, suggesting that the tracheal epithelium is the source of Wnt signals that induces Tbx4 in tracheal mesenchyme.

In vitro, Kishimoto et al. attempt to use this knowledge to differentiate human and mouse embryonic stem cells into putative tracheal mesenchyme cells. They suggest that exposure of embryonic stem cell-derived lateral plate mesoderm to Bmp and Wnt (and, in the case of human, Hedgehog) induces upregulation of Tbx4.

Major comments:

1. The hypothesis that bi-directional Wnt signaling between epithelium and mesenchyme is required for tracheal mesenchyme specification could be better substantiated. What are the Wnt ligands that the epithelium produces in order to instruct tracheal mesenchyme specification? The only evidence that the authors have to document the presence and importance of epithelium-derived Wnt signals at this point is drawn from the phenotype of their Shh-Cre;Wls(fl/fl) mice. But the authors would lend confidence to their hypothesis if they could show that the epithelium actually expresses Wnt ligand(s). Based on the authors' *in situ* staining, Wnt2 is unlikely to be the epithelium-derived Wnt ligand.
2. The authors suggest that the tracheal mesenchyme responds to Wnt signaling, because it is positive for a Lef1-GFP reporter. Although this is widely used in the field, it is an artificial reporter where Lef1 binding sites were fused to GFP (Currier et al., 2010; Genesis) and could thus be subject to technical artifacts. The authors should confirm that tracheal mesenchyme expresses endogenous Wnt target genes like Axin2, Lgr5 and/or Sp5, which should be quite straightforward using *in situ* hybridization and not require much additional work.
3. Do epithelium-derived Wnt signals directly or indirectly upregulate Tbx4 in the tracheal mesenchyme? For instance, are there Tcf/Lef binding sites in the Tbx4 promoter region and/or enhancer regions? It should be noted that Tbx3 can be a direct Wnt target gene (Renard et al., 2007; Cancer Research), so the authors could strengthen their hypothesis by demonstrating that Tbx4 is likewise a direct Wnt target gene that responds to epithelium-derived Wnt signals.
4. Timing: what is the chronological sequence of the bi-directional Wnt signaling between epithelium and mesenchyme? Based on the authors' Lef1-GFP reporter staining, it seems that at E9.5, the tracheal epithelium responds to Wnt, but only later (at E10.5) does the tracheal mesenchyme overtly respond to Wnt (Fig. 2). This makes it less likely that epithelium-derived Wnt signal is the critical, early signal inducing Tbx4 in tracheal mesenchyme, since Tbx4 is already

expressed in tracheal mesenchyme by E9.0 (Arora et al., 2012; PLoS Genetics). If Wnt is only active by E10.5, this suggests that Wnt is required for the maintenance (not specification) of tracheal mesenchyme, which is an interesting—but very different—conclusion.

5. Embryonic stem cell differentiation: first, for the mouse embryonic stem cell differentiation experiments, the authors should assess expression of both Tbx4 and Nkx6.1 (the latter of which they identify as a novel marker of tracheal mesenchyme); they only examine Tbx4 expression. Second, could the authors perform two-color in situ staining of mouse embryos to confirm that co-expression of Tbx4 and Nkx6.1 specifically marks tracheal mesenchyme? This is an important point, because if they can prove this, it would be a big boost to these experiments (validating the specificity of the markers examined), lending confidence that any in vitro-generated Tbx4+ Nkx6.1+ cells likely correspond to tracheal mesenchyme. Third, for the human embryonic stem cell differentiation experiments, the authors seem to largely use qPCR. They should co-stain for both Tbx4 and Nkx6.1 proteins (of note, there are good, well-validated Nkx6.1 antibodies for human cells; Schulz et al., 2012; PLoS ONE). Fourth, for both human and mouse embryonic experiments, it is important for the authors to quantify the purity of their cultures and to detail what percentage of cells are Tbx4+ Nkx6.1+. For instance, the Foxf1 and Tbx4 co-staining in Fig. 4c gives the impression that not many cells are double-positive. Fifth, do the in vitro-generated putative tracheal mesenchyme cells express key signaling molecules such as Wnt2 and Wnt2b, which are important for respiratory epithelium-mesenchyme communication (Goss et al., 2009; Dev Cell)?

Minor comments:

1. In Fig. 1b right hand panel, what are the bilateral spots of Tbx4 expression flanking the fused tracheal-esophageal tube? Are those the somites?
2. Fig. 2a,b: Very interestingly, at E9.5 it seems that Wnt signaling (as assayed by the Lef1-GFP reporter) is active in the ventral domain of the foregut endoderm (as expected). However, then at E10.5 and E11.5, does the overall level of Wnt signaling stay the same or change in the tracheal tube relative to E9.5? The different panels in Fig. 2a,b give different impressions about the extent of Lef1-GFP staining at E10.5 and E11.5.
3. The authors write “ShhCre, Ctnnb1flox/flox embryos did not express Tbx4, suggesting the activation of endodermal Wnt signaling, but not Nkx2.1, is required for following mesodermal Tbx4 expression” – although it is very unlikely, it is formally possible that Ctnnb1 deletion in other Shh+ lineages besides the endoderm (e.g., notochord, ventral somites, and ventral neural tube) may have led to non-cell autonomous effects on the Tbx4+ mesenchyme. Perhaps this should be mentioned briefly, unless the authors use another genetic driver that is more specific to the endoderm in order to ablate Ctnnb1.
4. “we conclude that mesodermal Wnt2 activates endodermal canonical Wnt signaling to express Nkx2.1 and Wnt ligands individually” – what does “individually” mean here in this context?
5. In Fig. 4 and Fig. S3 qPCR plots, what is “relative mRNA expression” (y-axis) relative to? Undifferentiated embryonic stem cells? If so, the fold-upregulation for Nkx6.1 mRNA is quite modest in some experiments. Immunostaining would be important to document the presence or absence of Nkx6.1 protein.
6. This is a minor and perhaps semantic point, but I thought the title of the manuscript (“Induction of tracheal mesoderm and chondrocyte from pluripotent stem cells in mouse and human”) did not capture the main essence of their advance, which is the idea that bi-directional Wnt signaling between tracheal epithelium and mesenchyme is important to confer tracheal-specific identity upon these mesenchyme cells. This notion of bi-directional Wnt signaling is quite exciting, and to me, the embryonic stem cell differentiation experiments are just one set of assays to test this hypothesis.

Reviewer #2:

Remarks to the Author:

In this short study Kishimoto et al. established a protocol for inducing tracheal/lung mesoderm and

differentiated mesenchymal tissue from pluripotent cells. In addition, they use the system to identify developmental signaling pathways coordinating mesodermal development. The findings support that mesodermal Wnts (possibly Wnt2) activates endodermal canonical Wnt signaling to initiate the expression of Nkx2.1 and Wnt ligands individually. These Wnt ligands then induce canonical Wnt signaling to initiate Tbx4 expression in the mesodermal progenitor cells. More importantly, the novel part of the study is that the authors used knowledge to generate tracheal cartilage and smooth muscle cells from both mouse and human cells. These findings are interesting, providing an innovative protocol to generate tracheal mesenchymal lineages for future studies. A few issues need to be addressed.

1. The study has used Tbx4 as an important marker for tracheal mesenchymal progenitor. Although the authors briefly mentioned Tbx4 in tracheal development, it will be necessary to give more details on the study by Aroar et al. regarding the phenotypes of Tbx4, Tbx5?
2. Formation of lung buds in Dermo1cre;cateninloxp/loxp mutants is not unpredicted. Multiple studies including by (Jiang et al., Development 2013 Sep;140(17):3589-94) have reported lung formation in the mutants.
3. It is interesting that the authors established seemingly successful protocol to generate tracheal mesenchymal lineages from mouse and human ESCs. These studies can be bolstered by more marker staining in addition to qPCR. These markers should be specific for smooth muscles and cartilage.

Jianwen Que

Reviewer #3:

Remarks to the Author:

In the submitted manuscript, Kishimoto et al unravel several new aspects of tracheal development and the interplay of endoderm to mesoderm in the formation of complex structures. The authors in vivo data documents a simple signaling interplay between the future respiratory endoderm and the surrounding splanchnic LPM that ultimately results in tracheal patterning. The authors then apply their newly found signaling interaction and individual marker genes into in vitro systems towards programming tracheal fates.

Overall, the in vivo work is (with one criticism on the used beta-catenin allele) nicely executed and documented, and provides novel insights into the endoderm-mesoderm interplay in a so-far not extensively studied developmental structure. The in vitro work, in contrast, seems rather preliminary in the manuscript's current form.

* Major:

- the authors use the classic Kemler allele of beta-catenin to perform their Cre/lox loss-of-function studies. However, this particular allele abolishes all beta-catenin, affecting both its Wnt function and its cell adhesion contribution. Other alleles, such as used in Valenta et al. 2011 (doi:0.1101/gad.181289.111) enable Wnt-only perturbations. While certainly too late for the authors to start using another allele, the text should mention that the tested Wnt association also affects cell adhesion in the Dermo1-Cre lineage, which could possibly add to the observed phenotypes, i.e. Tbx4 loss.

- The authors describe phenotypes affecting endoderm and splanchnic LPM defects, which are complex to the uninitiated reader if the nomenclature used is not consistent. The authors are encouraged to homogenize their use of tissue descriptions and to define structures more concisely, i.e. trachea mesenchymal development.

- The use of FoxF1 as key marker for the authors' in vitro programming into putative LPM is understandable and their approach is reasonable, but it seems rather superficial to define an entire mesodermal lineage. Did the authors check other markers for LPM in general, i.e. HoxB6, Prrx1,

among others? What is the percentage of reprogrammed cells at the individual steps that follow the authors' expected trajectory? The authors do see BMP4 induced, which is also an LPM-expressed gene, which should be emphasized.

- A key issue is the authors' claim that they programmed cells in vitro into tracheal lineages. This claim is only substantiated by correlative analysis of a handful of marker genes, including Tbx4, Sox9, and Nkx6.1. The authors are strongly encouraged to perform a transcriptome analysis on the resulting cells, or scRNA-seq to compare the transcriptional fingerprint of their resulting cells to actual tracheal lineages, as available as reference data with the back-to-back submitted manuscript.

* Minor:

- the abstract would benefit from more concise sentences to better convey the covered work.
- p4, l69: "splanchnic mesoderm" is indeed LPM, but more specifically the ventral fold of the LPM

Author's response to the reviewers

We thank the three reviewers for valuable comments to our manuscript. We addressed all concerns raised by reviewers. Our answers for the reviewer's comments are highlighted by blue color (Please see below).

Reviewers' comments:

Reviewer #1 (Remarks to the Author):

Development of endodermal tissues such as the tracheal epithelium critically depends on surrounding mesenchymal tissues that provide critical signals (e.g., Goss et al., 2009; Dev Cell) as well as mechanical forces (e.g., Kishimoto et al., 2018; Nature Communications). While the mesenchymal tissues that surround these various endodermal tissues were once regarded as rather similar, it is now known that they are quite different from one another: for instance, the mesenchyme surrounding the respiratory tract is unique, as it specifically expresses Tbx4 from E9.0-onwards in the mouse embryo (Arora et al., 2012; PLoS Genetics). While it has been partially investigated how tracheal epithelium progenitors are specified, it has remained relatively unknown how the surrounding tracheal mesenchyme is specified. Here, Kishimoto et al. address this important issue.

Kishimoto et al. perform both in vivo and in vitro experiments to address this question. In short, the authors find the tracheal mesenchyme fails to express Tbx4 if the ability of tracheal epithelium (Shh-Cre;Ctnnb1[fl/fl]) or tracheal mesenchyme (Dermo-Cre;Ctnnb1[fl/fl]) to respond to Wnt is impaired. The fact that a similar phenotype is seen when Ctnnb1 is deleted in either the tracheal epithelium or mesenchyme is quite striking, and suggests that bidirectional Wnt-mediated communication between the two tissues is required for tracheal mesenchyme specification. The authors further show that impairing the ability of the tracheal epithelium to secrete Wnt (Shh-Cre;Wls[fl/fl]) leads to the same phenotype, suggesting that the tracheal epithelium is the source of Wnt signals that induces Tbx4 in tracheal mesenchyme. In vitro, Kishimoto et al. attempt to use this knowledge to differentiate human and mouse embryonic stem cells into putative tracheal mesenchyme cells. They suggest that exposure of embryonic stem cell-derived lateral plate mesoderm to Bmp and Wnt (and, in the case of human, Hedgehog) induces upregulation of Tbx4.

Major comments:

1. The hypothesis that bi-directional Wnt signaling between epithelium and mesenchyme is required for tracheal mesenchyme specification could be better substantiated. What are the Wnt ligands that the epithelium produces in order to instruct tracheal mesenchyme specification? The only evidence that the authors have to document the presence and importance of epithelium-derived Wnt signals at this point is drawn from the phenotype of their Shh-Cre;Wls(fl/fl) mice. But the authors would lend confidence to their hypothesis if they could show that the epithelium actually expresses Wnt ligand(s). Based on the authors' in situ staining, Wnt2 is unlikely to be the epithelium-derived Wnt ligand.

We would like to thank the reviewer for this valuable comment. Past studies have shown that several Wnt ligands are expressed in the respiratory endoderm of mouse embryos. For instance, Jiang et al.

(Development, 2013) reported that Wnt3a, Wnt5a, Wnt6, Wnt7b, Wnt11 and Wnt16 are expressed in the endoderm at E11.5. Furthermore, Snitow et al. (Developmental Biology, 2015) have shown the expression of Wnt4, 5a, 7b, and 11 in endoderm at E11.5 and/or E13.5. However, it was unclear which ligands are expressed in endoderm at E10.5, prior to mesoderm specification. Current single cell RNA-seq data have shown the presence of several Wnt ligands including Wnt4, 5a, 5b, 6, 7b in the respiratory endoderm of mouse embryo at E9.5 (Han et al., back-to-back). We performed *in situ* hybridization (ISH) against these Wnt ligands at E10.5 (see below). Wnt4 was expressed in esophageal mesoderm and barely detected in tracheal endoderm. Wnt5a, 5b and 6 were detected in both epithelium and mesenchyme of the trachea. Wnt7b was abundantly expressed in tracheal endoderm, suggesting that Wnt7b might be responsible for the following induction of mesodermal Tbx4 expression. We added these data as new Figure 3e and Supplementary Figure S4, and also describe them in the manuscript.

2. The authors suggest that the tracheal mesenchyme responds to Wnt signaling, because it is positive for a Lef1-GFP reporter. Although this is widely used in the field, it is an artificial reporter where Lef1 binding sites were fused to GFP (Currier et al., 2010; Genesis) and could thus be subject to technical artifacts. The authors should confirm that tracheal mesenchyme expresses endogenous Wnt target genes like Axin2, Lgr5 and/or Sp5, which should be quite straightforward using *in situ* hybridization and not require much additional work.

We agree with the reviewer's comment. As the reviewer mentioned, there is a possibility that the signaling of Lef1-reporter mouse includes technical artifacts. Supporting our finding, however, is that Axin2-LacZ mouse also shows a similar reporter expression pattern to the LEF1-reporter line in the tracheal tissue at E10.5 (Woo et al. 2011 PLOS ONE, see below).

Axin2-LacZ (E10.5)

(Woo et al 2011)

Furthermore, we conducted RNAscope *in situ* hybridization experiment against Axin2, an endogenous canonical Wnt target, to confirm the activation of Wnt signaling in mesoderm. Consistent with the Lef1- and Axin2-reporter mouse experiments, Axin2 was expressed in tracheal endoderm and mesoderm at E10.5. We added this data as new Figure 2c and mentioned this in the manuscript.

3. Do epithelium-derived Wnt signals directly or indirectly upregulate Tbx4 in the tracheal mesenchyme? For instance, are there Tcf/Lef binding sites in the Tbx4 promoter region and/or enhancer regions? It should be noted that Tbx3 can be a direct Wnt target gene (Renard et al., 2007; Cancer Research), so the authors could strengthen their hypothesis by demonstrating that Tbx4 is likewise a direct Wnt target gene that responds to epithelium-derived Wnt signals.

Following the reviewer's suggestion, we explored T cell factor/lymphoid enhancer factor (Tcf/Lef) binding sites on the enhancer region of the mouse Tbx4 gene, particularly lung mesenchyme enhancer, Tbx4-LME (Menke et al Development 2008, Zhang et al BMC Biol 2013). We searched for the Tcf/Lef binding sequence by using the UCSC Genome Browser (<https://genome.ucsc.edu/>) and JASPAR (<http://jaspar.genereg.net/>) in this region. There were five repeats of Tcf/Lef binding sequences, which are well conserved between vertebrates except for fishes. These sequences have been identified as active cis-regulatory regions for H3K27Ac, H3K4me1 and p300 by ChIP-seq and by chromatin accessibility.

To further validate the response of these sequences to Wnt signaling, we established a new luciferase assay with mES-derived trachea mesodermal cells to examine enhancer activity in differentiating trachea mesodermal cells. The treatment of CHIR99021 enhanced Tbx4-LME reporter activity but not the mutated Tcf/Lef reporter. These results indicate that multiple Tcf/Lef binding sequences in Tbx4-LME are critical for Tbx4's response to canonical Wnt signaling during trachea mesodermal development. We added these data as new Figure 2f, 4f, and 4g, and mentioned this in the text.

Mouse Tbx4 lung mesenchyme enhancer

Luciferase reporter assay

4. Timing: what is the chronological sequence of the bi-directional Wnt signaling between epithelium and mesenchyme? Based on the authors' Lef1-GFP reporter staining, it seems that at E9.5, the tracheal epithelium responds to Wnt, but only later (at E10.5) does the tracheal mesenchyme overtly respond to Wnt (Fig. 2). This makes it less likely that epithelium-derived Wnt signal is the critical, early signal inducing Tbx4 in tracheal mesenchyme, since Tbx4 is already expressed in tracheal mesenchyme by E9.0 (Arora et al., 2012; PLoS Genetics). If Wnt is only active by E10.5, this suggests that Wnt is required

for the maintenance (not specification) of tracheal mesenchyme, which is an interesting—but very different—conclusion.

We apologize for the unclear description about the chronological sequence of the bi-directional Wnt signaling. As the reviewer mentioned, Arora et al. described “*Tbx4* expression is detected in the lung buds when they first appear a few hours later at E9.25 (28 somites),” but they did not mention anything about tracheal tissue. In addition, in Figure 1E of their paper, there is actually a bi-lateral domain of *Tbx4* expression in the lung bud region, but it is hard to see *Tbx4* expression in the putative trachea region. In order to clarify the chronological sequence of *Tbx4* expression in tracheal mesoderm at E9.5 to E10.5, we performed RNAscope *in situ* hybridization for *Tbx4*. At E9.5, *Tbx4* mRNA was detected in the mesoderm of the lung buds as previously reported, but not in the tracheal mesoderm. After tracheal-esophageal segregation at E10.5, both tracheal and lung mesoderm express *Tbx4*. We added these data as Supplementary Figure S1 and added a description in the revised manuscript.

5. Embryonic stem cell differentiation:

First, for the mouse embryonic stem cell differentiation experiments, the authors should assess expression of both *Tbx4* and *Nkx6.1* (the latter of which they identify as a novel marker of tracheal mesenchyme); they only examine *Tbx4* expression.

Following the reviewer’s suggestion, we tried double immunostaining against *Tbx4* and *Nkx6.1* on mESC-derived trachea mesenchymal cells. We tested two *Nkx6.1* antibodies (AF5857, R&D systems and F55A12, DSHB), but neither of these worked for detecting *Nkx6.1* even though *Nkx6.1* mRNA was detected in qRT-PCR at day 6 (data not shown). Thus, instead of detecting for *Nkx6.1* directly, we used the luciferase assay to demonstrate that *Tbx4*-LME drives *Tbx4* expression via canonical Wnt signaling in the mESC-derived trachea mesenchymal cells. This finding indicates that the induced trachea mesenchymal cells express a unique transcriptional feature of proper *in vivo* trachea mesenchymal cells. We added these data as new Figure 4f and 4g, and described them in the revised manuscript.

Second, could the authors perform two-color *in situ* staining of mouse embryos to confirm that co-expression of Tbx4 and Nkx6.1 specifically marks tracheal mesenchyme? This is an important point, because if they can prove this, it would be a big boost to these experiments (validating the specificity of the markers examined), lending confidence that any *in vitro*-generated Tbx4+ Nkx6.1+ cells likely correspond to tracheal mesenchyme.

Following this suggestion, we performed co-immunostaining for Tbx4/Nkx6.1 with mouse embryos instead of two-color *in situ* staining. Tbx4 was expressed in the entire tracheal mesoderm, but not in the esophagus. In contrast, Nkx6.1 was expressed in the dorsal tracheal mesoderm and esophageal mesoderm. Interestingly, Nkx6.1 expression was not detected in the ventral half of the tracheal mesoderm from E10.5 to E14.5. We thus defined three subtypes of mesoderm based on observations of the combination of Tbx4 and Nkx6.1 expression (i.e. Tbx4+/Nkx6.1-; ventral tracheal mesoderm, Tbx4+/Nkx6.1+; dorsal tracheal mesoderm, Tbx4-/Nkx6.1+; esophageal mesoderm).

We added these data as Supplementary Figure S5 and also mentioned them in the text.

Third, for the human embryonic stem cell differentiation experiments, the authors seem to largely use qPCR. They should co-stain for both Tbx4 and Nkx6.1 proteins (of note, there are good, well-validated Nkx6.1 antibodies for human cells; Schulz et al., 2012; PLoS ONE).

Thank you for the suggestion. We stained for TBX4 and NKX6.1 in hESC-derived trachea mesodermal cells to determine the ratio of double positive cells. Because Tbx4/Nkx6.1 double positive mesodermal cells in mouse embryos define the subtypes of trachea mesodermal cells, we counted these populations. At day 10, 30.3% of total cells expressed both TBX4 and NKX6.1, and 18.0% of the total cells expressed TBX4 but not Nkx6.1. These data indicate that the half of total cells were successfully differentiated into trachea mesodermal cells and that at least two kinds of tracheal mesoderm are present in our protocol. We added these data on new Figure 5f, 5g and describe them in the manuscript.

Fourth, for both human and mouse embryonic experiments, it is important for the authors to quantify the purity of their cultures and to detail what percentage of cells are Tbx4⁺ Nkx6.1⁺. For instance, the Foxf1 and Tbx4 co-staining in Fig. 4c gives the impression that not many cells are double-positive.

As the reviewer mentioned, we counted the number of differentiated cells based on marker expression as described above.

As for mESCs, Foxf1 and Gata4 staining showed that the most of cells (88.6% positive for Foxf1⁺) were differentiated into lateral plate mesoderm by day 5. After induction of tracheal mesoderm lineages, Tbx4⁺/Foxf1⁺ double positive cells appear in 89% of the cells by day 6, although we failed to detect Nkx6.1 protein in mouse trachea mesodermal cells as described above.

As for hESCs, a large majority of LPM cells were FOXF1⁺ and GATA4⁺ (95% positive for FOXF1) at day 2. After tracheal induction, 83% of the cells were TBX4⁺/FOXF1⁺ double positive trachea mesodermal cells at day 5. Furthermore, we detected 30% TBX4⁺/NKX6.1⁺ dorsal trachea mesodermal cells and 18% TBX4⁺/NKX6.1⁻ ventral trachea mesodermal cells at day 10.

We added these quantifications to Figures 4b, 4d, 5b, 5d and 5g, and also describe them in the revised manuscript.

Differentiation from mESC

Differentiation from hESC

Fifth, do the in vitro-generated putative tracheal mesenchyme cells express key signaling molecules such as Wnt2 and Wnt2b, which are important for respiratory epithelium-mesenchyme communication (Goss et al., 2009; Dev Cell)?

Mouse genetics have revealed that Wnt2 and Bmp are functionally essential for trachea development, but Wnt2b-null embryos showed no distinguishable phenotype (Goss et al. 2009 Dev Cell, Goss et al. 2011 Dev Biol). Therefore, we performed qRT-PCR analysis for Wnt2 and Bmp4. Both LPM and tracheal mesoderm expressed these growth factors in human and mouse. We added qRT-PCR data for Wnt2 and Bmp4 as new Figure 4e and 5e, and described them in the revised manuscript.

Minor comments:

1. In Fig. 1b right hand panel, what are the bilateral spots of Tbx4 expression flanking the fused tracheal-esophageal tube? Are those the somites?

These are non-specific background signals from blood cells in the dorsal aorta. We mention this in the figure legend.

2. Fig. 2a,b: Very interestingly, at E9.5 it seems that Wnt signaling (as assayed by the Lef1-GFP reporter) is active in the ventral domain of the foregut endoderm (as expected). However, then at E10.5 and E11.5, does the overall level of Wnt signaling stay the same or change in the tracheal tube relative to E9.5? The different panels in Fig. 2a,b give different impressions about the extent of Lef1-GFP staining at E10.5 and E11.5.

Thank you for the comment. We have added a more detailed explanation about the spatiotemporal changes of Wnt activation during segregation. At E9.5, Wnt signaling was mainly activated in the ventral endoderm. After tracheal-esophageal segregation at E10.5, Wnt signaling was temporally downregulated in the endoderm compared to that at E9.5. By contrast, mesodermal Wnt signaling was activated after segregation at E10.5. Both Lef1 reporter and Axin2 expression demonstrated that Wnt signaling activation is higher in mesoderm

than endoderm at E10.5. We added the data of Axin2 staining to new Fig. 2c and described it in the manuscript.

3. The authors write “ShhCre, Ctnnb1flox/flox embryos did not express Tbx4, suggesting the activation of endodermal Wnt signaling, but not Nkx2.1, is required for following mesodermal Tbx4 expression” – although it is very unlikely, it is formally possible that Ctnnb1 deletion in other Shh+ lineages besides the endoderm (e.g., notochord, ventral somites, and ventral neural tube) may have led to non-cell autonomous effects on the Tbx4+ mesenchyme. Perhaps this should be mentioned briefly, unless the authors use another genetic driver that is more specific to the endoderm in order to ablate Ctnnb1.

We understand the reviewer’s concern, and mention this possibility in the revised manuscript.

4. “we conclude that mesodermal Wnt2 activates endodermal canonical Wnt signaling to express Nkx2.1 and Wnt ligands individually” – what does “individually” mean here in this context?

We have rewritten this sentence as:

“We conclude that mesodermal Wnt2 activates endodermal canonical Wnt signaling which activates endodermal Wnt ligand expression independent of Nkx2.1.”

5. In Fig. 4 and Fig. S3 qPCR plots, what is “relative mRNA expression” (y-axis) relative to? Undifferentiated embryonic stem cells? If so, the fold-upregulation for Nkx6.1 mRNA is quite modest in some experiments. Immunostaining would be important to document the presence or absence of Nkx6.1 protein.

We agree that immunostaining is the better way to document the presence of Nkx6.1. We performed immunocytochemistry against NKX6.1 in addition to qRT-PCR. We added immunostaining data to new Figure 5f and describe this in the manuscript.

6. This is a minor and perhaps semantic point, but I thought the title of the manuscript (“Induction of tracheal mesoderm and chondrocyte from pluripotent stem cells in mouse and human”) did not capture the main essence of their advance, which is the idea that bi-directional Wnt signaling between tracheal epithelium and mesenchyme is important to confer tracheal-specific identity upon these mesenchyme cells. This notion of bi-directional Wnt signaling is quite exciting, and to me, the embryonic stem cell differentiation experiments are just one set of assays to test this hypothesis.

This is a very important suggestion, which we agree with. We have changed the title to “Bidirectional Wnt signaling between endoderm and mesoderm confer tracheal identity in mouse and human.”

--

Reviewer #2 (Remarks to the Author):

In this short study Kishimoto et al. established a protocol for inducing tracheal/lung mesoderm and differentiated mesenchymal tissue from pluripotent cells. In addition, they use the system to identify developmental signaling pathways coordinating mesodermal development. The findings support that mesodermal Wnts (possibly Wnt2) activates endodermal canonical Wnt signaling to initiate the expression of Nkx2.1 and Wnt ligands individually. These Wnt ligands then induce canonical Wnt signaling to initiate Tbx4 expression in the mesodermal progenitor cells. More importantly, the novel part of the study is that the authors used knowledge to generate tracheal cartilage and smooth muscle cells from both mouse and human cells. These findings are interesting, providing an innovative protocol to generate tracheal mesenchymal lineages for future studies. A few issues need to be addressed.

1. The study has used Tbx4 as an important marker for tracheal mesenchymal progenitor. Although the authors briefly mentioned Tbx4 in tracheal development, it will be necessary to give more details on the study by Arora et al. regarding the phenotypes of Tbx4, Tbx5?

We have added description about the function of Tbx4/5 genes on trachea/lung development based on the report by Arora et al. In brief, Tbx4 and Tbx5 cooperate to steer normal trachea development. Both genes are required for trachea mesodermal development, especially for cartilage and smooth muscle differentiation and morphogenesis. Tbx4, 5 double mutants show the tracheal stenotic phenotype.

2. Formation of lung buds in *Dermo1^{Cre};catenin^{loxp/loxp}* mutants is not unpredicted. Multiple studies including by (Jiang et al., *Development* 2013 Sep;140(17):3589-94) have reported lung formation in the mutants.

We apologize for omitting this important paper. As mentioned by the reviewer, Jiang et al. (*Development* 2013) have already reported the presence of lung buds in *Dermo1^{Cre}, β-catenin^{flox/flox}* mice. We mention this in the text and have added this paper in the reference list.

3. It is interesting that the authors established seemingly successful protocol to generate tracheal mesenchymal lineages from mouse and human ESCs. These studies can be bolstered by more marker staining in addition to qPCR. These markers should be specific for smooth muscles and cartilage.

We agree with the suggestions. We tested additional markers to verify differentiation of lateral plate mesoderm, trachea mesenchymal cells, chondrocytes and smooth muscle cells. We examined the expression of markers shown below by ICC and qRT-PCR,

Lateral plate mesoderm: *Foxf1, Hoxb6, Prrx1, Gata4, Bmp4*

Respiratory mesenchyme: *Tbx4/5, Wnt2, Bmp4, Nkx6.1.*

Chondrocyte: *Sox9, Col2a1, Acan. Sox5/6, Epyc*

Smooth muscle: *Acta2, Tagln, Col1a1.*

All lateral plate mesoderm markers showed obvious elevation at appropriate stages (day 5 mouse LPM and day 2 human LPM). In tracheal mesoderm, we confirmed Tbx4, Tbx5, Wnt2, Bmp4 and Nkx6.1 expression. Immunostaining also showed Tbx4⁺/Foxf1⁺ tracheal mesoderm (mouse 89%; human 83%).

In human tracheal mesoderm, we detected subtypes of trachea mesodermal cells as the combination of TBX4 and NKX6.1 expression. See our answer for major comment #5 of Reviewer #1 regarding subtypes of trachea mesodermal cells.

At day 12 of mESC and day 10 of hESC cultures, the expression of chondrogenic markers and smooth muscle cell markers were detected.

We added these data in Fig. 4, 5, S6 and mentioned them in the manuscript.

Reviewer #3 (Remarks to the Author):

In the submitted manuscript, Kishimoto et al unravel several new aspects of tracheal development and the interplay of endoderm to mesoderm in the formation of complex structures. The authors in vivo data documents a simple signaling interplay between the future respiratory endoderm and the surrounding splanchnic LPM that ultimately results in tracheal patterning. The authors then apply their newly found signaling interaction and individual marker genes into in vitro systems towards programming tracheal fates. Overall, the in vivo work is (with one criticism on the used beta-catenin allele) nicely executed and documented, and provides novel insights into the endoderm-mesoderm interplay in a so-far not extensively studied developmental structure. The in vitro work, in contrast, seems rather preliminary in the manuscript's current form.

* Major:

- the authors use the classic Kemler allele of beta-catenin to perform their Cre/lox loss-of-function studies. However, this particular allele abolishes all beta-catenin, affecting both its Wnt function and its cell adhesion contribution. Other alleles, such perturbations. While certainly too late for the authors to start using another allele, the text should mention that the tested Wnt association also affects cell adhesion in the *Dermo1*-Cre lineage, which could possibly add to the observed phenotypes, i.e. *Tbx4* loss.

We agree with this point. It is undeniable that altered cell adhesion affects *Tbx4* expression in trachea mesenchymal cells. We mentioned the possibility that the allele abolishes all beta-catenin, affecting both its contribution to Wnt function and cell adhesion, and hence, may influence *Tbx4* expression. On the other hand, we showed that epithelial Wnt ligand inactivation with *Shh^{Cre}*, *Wls^{flox/flox}* embryos cause loss of *Tbx4* expression (Fig. 3). We also found that CHIR99021, a GSK3 β inhibitor, activates a lung mesenchyme enhancer of *Tbx4* in the mouse tracheal mesenchyme *in vitro*. These data imply that Wnt-mediated transcriptional regulation, but not cell adhesion, affects *Tbx4* expression. Supporting these findings, *Dermo1^{Cre}*, *Ctnnb1^{flox/flox}* embryos did not show a difference in the distribution of CDH2 as an adhesive molecule of tracheal mesenchyme. We added these data as new Figure 2f, 4f, g and Supplementary Figure S3, and mentioned these points in the text.

- The authors describe phenotypes affecting endoderm and splanchnic LPM defects, which are complex to the uninitiated reader if the nomenclature used is not consistent. The authors are encouraged to homogenize their use of tissue descriptions and to define structures more concisely, i.e. trachea mesenchymal development.

Thank you for the suggestion. We have carefully checked the manuscript and corrected nomenclature to be consistent.

- The use of FoxF1 as key marker for the authors' in vitro programming into putative LPM is understandable and their approach is reasonable, but it seems rather superficial to define an entire mesodermal lineage. Did the authors check other markers for LPM in general, i.e. HoxB6, Prrx1, among others? What is the percentage of reprogrammed cells at the individual steps that follow the authors' expected trajectory? The authors do see BMP4 induced, which is also an LPM-expressed gene, which should be emphasized.

Thank you for the suggestion. As we responded to Reviewer #2, we checked the expressions of the LPM markers Foxf1, Hoxb6, Prrx1, Gata4, and Bmp4 at appropriate stages. As seen above, we have conducted qRT-PCR for these marker expressions in hESC and mESC-driven LPM and detected clear elevation of all of these marker genes. Immunostaining for Foxf1 and Gata4 revealed that 88~95% of cells are differentiated into LPM. For respiratory fate, we examined Tbx4, Tbx5, Wnt2 and Nkx6.1 expression by qRT-PCR. These markers were upregulated after differentiation induction. Furthermore, immunostaining showed that 82%~89% of mESC and hESC-derived trachea mesodermal cells include Tbx4/Foxf1 double positive cells.

We added these data in new Figure 4 and 5, and described them in the manuscript.

- A key issue is the authors' claim that they programmed cells in vitro into tracheal lineages. This claim is only substantiated by correlative analysis of a handful of marker genes, including Tbx4, Sox9, and Nkx6.1. The authors are strongly encouraged to perform a transcriptome analysis on the resulting cells, or scRNA-seq to compare the transcriptional fingerprint of their resulting cells to actual tracheal lineages, as available as reference data with the back-to-back submitted manuscript.

We agree that various markers must be checked in differentiating mESC/hESCs. We checked the expression of numerous genes that are involved in respiratory mesoderm development (see below), instead of a comprehensive transcriptome analysis.

Lateral plate mesoderm: Foxf1, Hoxb6, Prrx1, Gata4, Bmp4.

Respiratory mesenchyme: Tbx4/5, Wnt2, Bmp4, Nkx6.1.

Chondrocyte: Sox9, Col2a1, Acan, Sox5, Sox6, Epyc

Smooth muscle: Acta2, Tagln, Col1a1.

As mentioned above, these markers displayed strong expression at appropriate stages of differentiation. We added these data in new Figure 4, 5 and S6 and mentioned them in the manuscript.

* Minor:

- the abstract would benefit from more concise sentences to better convey the covered work.

We have carefully revised our writing, and asked a native English speaker to check our English.

- p4, l69: "splanchnic mesoderm" is indeed LPM, but more specifically the ventral fold of the LPM

Thank you for the comment. We changed “splanchnic mesoderm” to “ventral fold of the LPM”.

Reviewers' Comments:

Reviewer #1:

Remarks to the Author:

In their revised manuscript, Kishimoto et al. provide additional data to address some of the reviewers' comments, including 1) *in situ* stains for Wnt ligands in the tracheal endoderm, 2) confirmation that tracheal mesenchyme responds to Wnt signaling (as shown by Axin2 expression), 3) clarification that while the tracheal mesenchyme is Tbx4+, only a subset is Nkx6.1+ *in vivo* and 4) further quantification of the purity of pluripotent stem cells (PSC)-derived mesenchymal populations (including, notably, staining for NKX6.1 protein expression). However, there are additional steps that the authors could take to strengthen their manuscript even further.

General comments:

1. Introduction: While the Introduction summarizes some of the past literature, the paper is not the easiest to read; this could be remedied if the authors concisely provided a clear model in the Introduction and then set out to outline what they need to prove. Providing a unifying model in the beginning would help the reader quite a bit. The authors could do a better job to more clearly lay out their argument for the reader, thus framing the study and what new findings the reader can expect. First, by way of background, it is known from past work that mesenchymal cells secrete signals (e.g., BMP and WNT) to induce respiratory endoderm identity. Here we propose that this communication is bidirectional, namely that endoderm cells also produce signals (e.g., WNT) that induce respiratory mesenchymal identity. Our model is that at E9.5, the tracheal/lung endoderm is specified (as shown by Nkx2.1 expression), which secretes Wnt ligands that act on the tracheal mesenchyme to induce Tbx4 at E10.5. To substantiate this model, here we need to prove the following key points: (1) tracheal endoderm secretes Wnt ligands; (2) tracheal mesenchyme responds to Wnt ligands; (3) tracheal mesenchyme must respond to Wnt signaling in order to induce Tbx4; (4) Tbx4 is probably a Wnt target gene.
2. Discussion: The discussion section is quite short, and could be expanded in order to better integrate current and past findings.
3. *In vivo* analyses: Other mesodermal or mesenchymal markers could have been analyzed in Dermo-Cre or Shh-Cre; β Catenin and Wls mutants. Throughout their paper, the authors principally analyze Tbx4 expression (with the exception of one supplementary figure looking at Cdh2) as a single marker for respiratory mesenchyme identity. The use of additional markers could strengthen their conclusions. Perhaps the authors would consider looking at proliferation and other mesodermal markers (Nkx6.1, FoxF1, Gata4 or other markers of region-specific mesenchyme?) to help to clarify their point? For instance, it would be illuminating if the authors found in their Wnt signaling-defective mutants that the surrounding mesenchymal cells still express pan-mesenchymal markers (e.g., Foxf1) but selectively lacked respiratory mesenchyme-specific markers (e.g., Tbx4). (For instance, in line 148, the authors write "These Wnt ligands then induce endodermal-to-mesodermal canonical Wnt signaling to initiate mesodermal Tbx4 expression" – it would be even better if the authors could write "to initiate respiratory mesenchyme-specific identity", as shown by additional markers, instead of just assessing Tbx4 expression.)
4. Tbx4 as Wnt target gene: Luciferase reporter assay in Fig. 4g shows a less than 2-fold increase in enhancer activity upon Wnt agonist treatment; the effect seems specific (not observed in Tcf/Lef mutants) but still fairly modest.

Minor comments:

1. Figures (for instance, some panels of Figs. 2, 4 and 5) are sometimes of lower quality, was this due to image compression during file upload?
2. Title ("Bidirectional Wnt signaling between endoderm and mesoderm confer tracheal identity in mouse and human"): The "bidirectional" concept is a nice touch, as it was shown in mouse, although the human PSC differentiation experiments do not have anything to do with "bidirectional" signaling, and therefore the authors should consider removing "human" from the title.
3. There are minor grammatical mistakes throughout the manuscript that should be corrected, like

“developmental signaling pathways coordinating the mesodermal development are still undefined” (pg. 3) and “The tracheal mesoderm originates from ventral fold of the lateral plate mesoderm” (pg. 3) and elsewhere throughout the paper

4. “At E9.5, Tbx4 is only detected in lung buds but not tracheal mesoderm (Supplementary Fig. S1)” (pg. 4) – Fig. S1 does not seem to show Tbx4 expression in the endoderm, maybe since the Nkx2.1 staining overpowers the Tbx4 staining in the superimposed images?

5. “At the same time as endodermal Nkx2.1 induction, tracheal/lung mesoderm is also defined, expressing Tbx4/5 by E10.5” (pg. 4) – Is this correct? I was under the impression that Nkx2.1 is induced at E9.5 in endoderm, and therefore if Tbx4 is induced in the surrounding mesenchyme at E10.5, it would appear that respiratory endoderm identity is induced 24 hours prior to respiratory mesenchymal identity?

6. Regarding the conclusion that endoderm-derived Wnt is dispensable for Tbx4 induction in the lung bud mesenchyme, Fig. S2b seems to nonetheless show that Tbx4 is decreased compared to control. Is this just an artifact of the section that was examined, or is there still a modest phenotype in the lung?

7. “These sequences are active cis-regulatory regions for H3K27Ac, H3K4me1 and p300 as determined by ChIP-seq and by chromatin accessibility” (pg. 6) – The authors should mention the precise developmental stage/tissue in which these ChIP-seq assays were performed and whether the biological contexts in which these assays were performed are relevant to respiratory mesenchyme. If an enhancer is H3K27Ac+ in a tissue that is not the respiratory mesenchyme, this might not be relevant.

8. Fig. 5i – figure itself should be labeled to clarify that the triangles indicate increasing CHIR (not just in the figure legend)

9. Fig. S7 – When the authors are testing the presence/absence of different signals or testing combinations thereof, what timeframe are the authors performing these signaling manipulations? The timing when these signals (e.g., BMP, HH and WNT) are provided is critical. For instance, in Fig. S7b, are the signals manipulated in the day 2-to-day 5 interval of “tracheal mesoderm” induction and then readout at day 5 (for part of the figure) and then manipulated in the day 5-to-day 10 interval and then readout at day 10 (for part of the figure)?

10. Fig. S7 is incredibly interesting, and readers may want to know what the relative contribution of BMP vs. HH vs. WNT is in inducing tracheal mesoderm fate. The authors show some, but not all, of the key combinations of signals: for instance, HH seems to increase TBX4 by ~2-fold, but WNT alone seems to substantially induce TBX4 already (Fig. S7c). What about if no WNT is added, or even a WNT inhibitor? If the authors still have cDNA from this qPCR experiment, it might be interesting to perform qPCR for additional pan-lateral mesoderm markers like FOXF1, HAND1 and GATA4. For instance, it could be that signal X alone is enough to induce pan-lateral mesoderm, but that signal X and Y together act to induce respiratory-specific mesoderm, which would be quite interesting and could corroborate the authors’ in vivo work.

Reviewer #2:

Remarks to the Author:

The authors have addressed all of this reviewer's concern.

Reviewer #3:

Remarks to the Author:

In the revised manuscript, Kishimoto and colleagues have performed a considerable amount of additional experiments and re-writing that in sum has much improved the manuscript. All added data is once more of high quality.

The addition of the Wnt in situs and Wnt signaling activity readouts, as requested by a fellow reviewer, added valuable reference data for these critical expression patterns.

The additional explanations mitigating the concerns with the used beta-catenin allele will also assist other investigators in the interpretation of findings using this mutant.

Minor comments:

1) line 29 in the abstract, the phrase on "downstream Wnt ligand" seems off and needs revisiting; similar for line 31 "Repopulating in vivo model".

2) The authors added a good amount of additional LPM markers, yet should also provide references justifying their use to support uninitiated readers as to why these markers are relevant. The LPM has been only vaguely defined in the literature in the past, and only recently several publications across models have provided a better context to this complex biology. The authors have ample room for adding more references.

3) Referencing of the co-submitted Han et al manuscript should be adapted depending on publication timeline and peer review outcome.

4) in Figure 2 e', can the authors possibly use the same color code in the schematic as in the 3D renderings in Figure 2 e?

5) in Figure 2 f, can the authors add a mention of the TCF/LEF consensus sequence that guided them to assign the green sequences as such putative binding sites?

Author's response to the reviewers

We thank again the reviewers for valuable comments to our manuscript. We addressed all concerns raised by the reviewers. Our answers for the reviewer's comments are highlighted by blue color (Please see below).

REVIEWER COMMENTS

Reviewer #1 (Remarks to the Author):

In their revised manuscript, Kishimoto et al. provide additional data to address some of the reviewers' comments, including 1) in situ stains for Wnt ligands in the tracheal endoderm, 2) confirmation that tracheal mesenchyme responds to Wnt signaling (as shown by Axin2 expression), 3) clarification that while the tracheal mesenchyme is Tbx4+, only a subset is Nkx6.1+ in vivo and 4) further quantification of the purity of pluripotent stem cells (PSC)-derived mesenchymal populations (including, notably, staining for NKX6.1 protein expression). However, there are additional steps that the authors could take to strengthen their manuscript even further.

General comments:

1. Introduction: While the Introduction summarizes some of the past literature, the paper is not the easiest to read; this could be remedied if the authors concisely provided a clear model in the Introduction and then set out to outline what they need to prove. Providing a unifying model in the beginning would help the reader quite a bit. The authors could do a better job to more clearly lay out their argument for the reader, thus framing the study and what new findings the reader can expect. First, by way of background, it is known from past work that mesenchymal cells secrete signals (e.g., BMP and WNT) to induce respiratory endoderm identity. Here we propose that this communication is bidirectional, namely that endoderm cells also produce signals (e.g., WNT) that induce respiratory mesenchymal identity. Our model is that at E9.5, the tracheal/lung endoderm is specified (as shown by Nkx2.1 expression), which secretes Wnt ligands that act on the tracheal mesenchyme to induce Tbx4 at E10.5. To substantiate this model, here we need to prove the following key points: (1) tracheal endoderm secretes Wnt ligands; (2) tracheal mesenchyme responds to Wnt ligands; (3) tracheal mesenchyme must respond to Wnt signaling in order to induce Tbx4; (4) Tbx4 is probably a Wnt target gene.

Thank you for good suggestions. We revised introduction and explained our model and key points of this paper following the suggestions.

2. Discussion: The discussion section is quite short, and could be expanded in order to better integrate current and past findings.

Thank you for comments. We revised and added some discussion about the difference between species, the role of Tbx4 on mesodermal development, and difference between the trachea and the lung.

3. In vivo analyses: Other mesodermal or mesenchymal markers could have been analyzed in Dermo-Cre or Shh-Cre; bCatenin and Wls mutants. Throughout their paper, the authors principally analyze Tbx4 expression (with the exception of one supplementary figure looking at Cdh2) as a single marker for respiratory mesenchyme identity. The use of additional markers could strengthen their conclusions. Perhaps the authors would consider looking at proliferation and other mesodermal markers (Nkx6.1, FoxF1, Gata4 or other markers of region-specific mesenchyme?) to help to clarify their point? For instance, it would be illuminating if the authors found in their Wnt signaling-defective mutants that the surrounding mesenchymal cells still express pan-mesenchymal markers (e.g., Foxf1) but selectively lacked respiratory mesenchyme-specific markers (e.g., Tbx4). (For instance, in line 148, the authors write “These Wnt ligands then induce endodermal-to-mesodermal canonical Wnt signaling to initiate mesodermal Tbx4 expression” – it would be even better if the authors could write “to initiate respiratory mesenchyme-specific identity”, as shown by additional markers, instead of just assessing Tbx4 expression.)

We performed immunostaining for Foxf1 in mutant embryos. Foxf1 was expressed in mesoderm of all Wnt mutant trachea. We added these data as Supplementary Figure 2a-c and, as the reviewer suggested, revised our text to “Wnt ligands induce endodermal-mesodermal canonical Wnt signaling to initiate respiratory mesenchyme-specific identity”.

4. Tbx4 as Wnt target gene: Luciferase reporter assay in Fig. 4g shows a less than 2-fold increase in enhancer activity upon Wnt agonist treatment; the effect seems specific (not observed in Tcf/Lef mutants) but still fairly modest.

As the reviewer mentioned, we observed only less than 2-fold increase in luciferase activity upon Wnt activator treatment. We spent significant amount of time to optimize the transfection of mESC undergoing differentiation during LPM (day 5 in the differentiation protocol). The fairly increase may be due to the low transfection efficiency, which was less than 15% (judged from EGFP-expression). However, this increase was statistically significant compared to control and Tbx4LME mutant as we described in the text.

Minor comments:

1. Figures (for instance, some panels of Figs. 2, 4 and 5) are sometimes are of lower quality, was this due to image compression during to file upload?

As the reviewer mentioned, some figures (Figs. 2e, 4d, 5d, 5f) look pixelized. We replaced these figures to higher quality figures.

2. Title (“Bidirectional Wnt signaling between endoderm and mesoderm confer tracheal identity in mouse and human”): The “bidirectional” concept is a nice touch, as it was shown in mouse, although the human PSC differentiation experiments do not have anything to do with “bidirectional” signaling, and therefore the authors should consider removing “human” from the title.

I'm afraid but understanding of human trachea development is one of important focus of our current report. If we remove “human” from our title, readers may assume that this research is limited in murine resources. To avoid such a misinterpretation, we would like to remove both mouse and human, like bellow

Bidirectional Wnt signaling between endoderm and mesoderm confers tracheal identity.

3. There are minor grammatical mistakes throughout the manuscript that should be corrected, like “developmental signaling pathways coordinating the mesodermal

development are still undefined” (pg. 3) and “The tracheal mesoderm originates from ventral fold of the lateral plate mesoderm” (pg. 3) and elsewhere throughout the paper

We carefully corrected grammatical mistakes in the manuscripts.

4. “At E9.5, Tbx4 is only detected in lung buds but not tracheal mesoderm (Supplementary Fig. S1)” (pg. 4) – Fig. S1 does not seem to show Tbx4 expression in the endoderm, maybe since the Nkx2.1 staining overpowers the Tbx4 staining in the superimposed images?

Sorry for our unclear description. Tbx4 is not expressed in endoderm at lung buds. We revised “lung buds” to “lung buds’ mesoderm”.

5. “At the same time as endodermal Nkx2.1 induction, tracheal/lung mesoderm is also defined, expressing Tbx4/5 by E10.5” (pg. 4) – Is this correct? I was under the impression that Nkx2.1 is induced at E9.5 in endoderm, and therefore if Tbx4 is induced in the surrounding mesenchyme at E10.5, it would appear that respiratory endoderm identity is induced 24 hours prior to respiratory mesenchymal identity?

We agree. We corrected the manuscript.

6. Regarding the conclusion that endoderm-derived Wnt is dispensable for Tbx4 induction in the lung bud mesenchyme, Fig. S2b seems to nonetheless show that Tbx4 is decreased compared to control. Is this just an artifact of the section that was examined, or is there still a modest phenotype in the lung?

As this reviewer indicates, we cannot exclude the possibility that endodermal Wls deletion slightly alters Tbx4 expression. We revised the text as bellow,

“Disruption of Wnt signaling in the mesoderm eliminate Tbx4 expression in the tracheal but still detectable in lung mesoderm, suggesting that Wnt-mediated mesodermal Tbx4 induction is a unique system in tracheal development but not lung development.

7. “These sequences are active cis-regulatory regions for H3K27Ac, H3K4me1 and p300

as determined by ChIP-seq and by chromatin accessibility” (pg. 6) – The authors should mention the precise developmental stage/tissue in which these ChIP-seq assays were performed and whether the biological contexts in which these assays were performed are relevant to respiratory mesenchyme. If an enhancer is H3K27Ac+ in a tissue that is not the respiratory mesenchyme, this might not be relevant.

We have described the developmental stages, tissue and ENCODE accession numbers for the ChIP-seq and ATAC-seq data shown in Figure 2f. These data came from E14.5/E15.5/PND0 lung in which the transcription of *Tbx4* gene is spatial and temporally active (Arora et al., 2012, Guo et al., 2019, Lüdtke et al., 2013, Zhang et al., 2013).

8. Fig. 5i – figure itself should be labeled to clarify that the triangles indicate increasing CHIR (not just in the figure legend)

We labeled the triangles with CHIR in the figure.

9. Fig. S7 – When the authors are testing the presence/absence of different signals or testing combinations thereof, what timeframe are the authors performing these signaling manipulations? The timing when these signals (e.g., BMP, HH and WNT) are provided is critical. For instance, in Fig. S7b, are the signals manipulated in the day 2-to-day 5 interval of “tracheal mesoderm” induction and then readout at day 5 (for part of the figure) and then manipulated in the day 5-to-day 10 interval and then readout at day 10 (for part of the figure)?

We tested the presence/absence of different signals from day2 lateral plate mesoderm in all experimental conditions. Therefore, day 5 samples were exposed to signals from day2 to day5, and day10 samples were exposed from day2 to day10. We described the time-course of the experiment in figure legends.

10. Fig. S7 is incredibly interesting, and readers may want to know what the relative contribution of BMP vs. HH vs. WNT is in inducing tracheal mesoderm fate. The authors show some, but not all, of the key combinations of signals: for instance, HH seems to increase *TBX4* by ~2-fold, but WNT alone seems to substantially induce *TBX4* already (Fig. S7c). What about if no WNT is added, or even a WNT inhibitor? If the authors still have cDNA from this qPCR experiment, it might be interesting to perform qPCR for additional pan-lateral mesoderm markers like *FOXF1*, *HAND1* and *GATA4*. For instance,

it could be that signal X alone is enough to induce pan-lateral mesoderm, but that signal X and Y together act to induce respiratory-specific mesoderm, which would be quite interesting and could corroborate the authors' in vivo work.

Thank you for being interested in our experiments. In terms of the presence/absence of Wnt experiments, we have shown the data in figure 5i, Supplementary figure 8e and 8f. Furthermore, as the reviewer suggested, we examined FOXF1 expression in the presence/absence of Wnt experiments. FOXF1 expression was decreased in dose-dependent manner of CHIR, but still expressed during differentiation. Because FOXF1 expression is downregulated in ventral respiratory mesoderm, this result also suggested the successful induction of respiratory mesoderm. Combining the data of FOXF1/TBX4 expression, HH/BMP was enough to induce LPM but HH/BMP/WNT was required for respiratory mesoderm induction. These results were consistent with in vivo work that *Foxf1* was retained but *Tbx4* was lost in Wnt mutant (eg; *Shh^{Cre}; Wls^{fllox/fllox}* and *Dermo1Cre; Ctnnb1^{fllox/fllox}*). These data suggested WNT signaling promotes respiratory fate in dose-dependent manner. We also tested the presence/absence of HH and BMP4 experiments as shown in Supplementary figure 8. Based on these data sets, HH/WNT/(BMP) was the most efficient combination to induce respiratory mesoderm. We added these data as Supplementary Figure 8g and mentioned in the manuscript.

--

Reviewer #2 (Remarks to the Author):

The authors have addressed all of this reviewer's concern.

--

Reviewer #3 (Remarks to the Author):

In the revised manuscript, Kishimoto and colleagues have performed a considerable amount of additional experiments and re-writing that in sum has much improved the manuscript. All added data is once more of high quality.

The addition of the Wnt in situs and Wnt signaling activity readouts, as requested by a fellow reviewer, added valuable reference data for these critical expression patterns.

The additional explanations mitigating the concerns with the used beta-catenin allele will also assist other investigators in the interpretation of findings using this mutant.

Minor comments:

1) line 29 in the abstract, the phrase on "downstream Wnt ligand" seems off and needs revisiting; similar for line 31 "Repopulating in vivo model".

We replaced "its downstream Wnt ligand" to "Wnt ligand secretion". Also, we removed 'Repopulating in vivo model'.

2) The authors added a good amount of additional LPM markers, yet should also provide references justifying their use to support uninitiated readers as to why these markers are relevant. The LPM has been only vaguely defined in the literature in the past, and only recently several publications across models have provided a better context to this complex biology. The authors have ample room for adding more references.

We added some critical references to rationalize the usage of these LPM marker.

3) Referencing of the co-submitted Han et al manuscript should be adapted depending on publication timeline and peer review outcome.

We already provided Han et al paper in the main text. We will be sure to add this paper in the list of references.

4) in Figure 2 e', can the authors possibly use the same color code in the schematic as in the 3D renderings in Figure 2 e?

Thank you for suggestion. We changed the color code in Figure 2e'.

5) in Figure 2 f, can the authors add a mention of the TCF/LEF consensus sequence that guided them to assign the green sequences as such putative binding sites?

We have added the consensus sequences below each putative binding site in Figure 2f.

Reviewers' Comments:

Reviewer #1:

Remarks to the Author:

Kishimoto et al. have satisfactorily addressed the majority of our concerns. It is intriguing that Tbx4 is specifically lost in the Wnt pathway mutant embryos, but that Foxf1 continues to be expressed, suggesting that respiratory mesenchyme identity is specifically lost upon Wnt ablation, but that mesenchymal cells persist. Also the expanded Discussion section does a good job of emphasizing some of the highlights of the authors' work, for instance the notion that the importance of HH signaling might differ between human and mouse, as well as the poorly understood differences between tracheal and lung development.

Some remaining minor comments:

1. Minor grammatical corrections could be made throughout. For instance: "The Tbx4 expression relies on endodermal Wnt activation and Wnt ligand secretion but independent of known Nkx2.1-mediated respiratory development" (pg. 2) should be corrected to "Tbx4 expression relies on endodermal Wnt activation and Wnt ligand secretion but is independent of known Nkx2.1-mediated respiratory development" ("mesenchymal" could also be added before "Tbx4 expression" to make it clear what is the subject being referred to by "Tbx4 expression"). Another phrase that could be corrected is "lung buds mesoderm" (Introduction) which should be corrected to "lung bud mesoderm"
2. References should be provided for the statement "These studies have demonstrated that mesodermal cells secrete growth factors (eg; Wnt and Bmp) to induce respiratory endoderm identity." (Introduction)
3. The order in which ideas are presented in the Introduction could be slightly tweaked. The 2nd paragraph of the Introduction discusses the bidirectional signaling inducing mesenchymal Tbx4 expression, but only in the 3rd paragraph is it explained that Tbx4 is a significant mesenchymal marker. The order of those paragraphs might be flipped to help the reader understand.
4. "Canonical Wnt signalling induces Tbx4 expression in the tracheal mesoderm but not in the lungs" (Discussion) – this is certainly a very interesting paragraph. However, it seems that it could benefit from a topic sentence introducing the scientific question, for instance, "Another question is how tracheal and lung are differentially specified, despite the strong commonalities between them", which would provide context for the reader to subsequently understand the paragraph's details.

Author's response to the reviewers; the final revisions for NCOMMS-1124663B

We thank again the reviewer for valuable comments to our manuscript. We addressed all concerns raised by the reviewers. Our answers for the reviewer's comments are highlighted by blue color (Please see below).

REVIEWERS' COMMENTS:

Reviewer #1 (Remarks to the Author):

Kishimoto et al. have satisfactorily addressed the majority of our concerns. It is intriguing that Tbx4 is specifically lost in the Wnt pathway mutant embryos, but that Foxf1 continues to be expressed, suggesting that respiratory mesenchyme identity is specifically lost upon Wnt ablation, but that mesenchymal cells persist. Also the expanded Discussion section does a good job of emphasizing some of the highlights of the authors' work, for instance the notion that the importance of HH signaling might differ between human and mouse, as well as the poorly understood differences between tracheal and lung development.

Some remaining minor comments:

1. Minor grammatical corrections could be made throughout. For instance: "The Tbx4 expression relies on endodermal Wnt activation and Wnt ligand secretion but independent of known Nkx2.1-mediated respiratory development" (pg. 2) should be corrected to "Tbx4 expression relies on endodermal Wnt activation and Wnt ligand secretion but is independent of known Nkx2.1-mediated respiratory development" ("mesenchymal" could also be added before "Tbx4 expression" to make it clear what is the subject being referred to by "Tbx4 expression"). Another phrase that could be corrected is "lung buds mesoderm" (Introduction) which should be corrected to "lung bud mesoderm"

Thank you for noticing the grammatical correction. We corrected the text.

2. References should be provided for the statement "These studies have demonstrated that mesodermal cells secrete growth factors (eg; Wnt and Bmp) to induce respiratory endoderm identity." (Introduction)

We added references.

3. The order in which ideas are presented in the Introduction could be slightly tweaked. The 2nd paragraph of the Introduction discusses the bidirectional signaling inducing mesenchymal Tbx4 expression, but only in the 3rd paragraph is it explained that Tbx4 is a significant mesenchymal marker. The order of those paragraphs might be flipped to help the reader understand.

We agreed. We flipped the order of the 2nd and 3rd paragraph in the introduction section.

4. "Canonical Wnt signalling induces Tbx4 expression in the tracheal mesoderm but not in the lungs" (Discussion) – this is certainly a very interesting paragraph. However, it seems that it could benefit from a topic sentence introducing the scientific question, for instance, "Another question is how tracheal and lung are differentially specified, despite the strong commonalities between them", which would provide context for the reader to subsequently understand the paragraph's details.

Yes, we provided this scientific question with the top sentence of this paragraph.